# Pten controls B-cell responsiveness and germinal center reaction by regulating the expression of IgD BCR

Corinna S Setz[1], Ahmad Khadour[1], Valerio Renna[1] (iD), Joseena Iype[1,2,†], Eva Gentner[1], Xiaocui He[2,‡], Moumita Datta[1], Marc Young[1], Lars Nitschke[3], Jürgen Wienands[4], Palash C Maity[1], Michael Reth[2] (iD) & Hassan Jumaa[1,*] (iD)

## Abstract

In contrast to other B-cell antigen receptor (BCR) classes, the function of IgD BCR on mature B cells remains largely elusive as mature B cells co-express IgM, which is sufficient for development, survival, and activation of B cells. Here, we show that IgD expression is regulated by the forkhead box transcription factor FoxO1, thereby shifting the responsiveness of mature B cells towards recognition of multivalent antigen. FoxO1 is repressed by phosphoinositide 3-kinase (PI3K) signaling and requires the lipid phosphatase Pten for its activation. Consequently, Pten-deficient B cells expressing knock-ins for BCR *heavy* and *light chain* genes are unable to upregulate IgD. Furthermore, in the presence of autoantigen, Pten-deficient B cells cannot eliminate the autoreactive BCR specificity by secondary *light chain* gene recombination. Instead, Pten-deficient B cells downregulate BCR expression and become unresponsive to further BCR-mediated stimulation. Notably, we observed a delayed germinal center (GC) reaction by IgD-deficient B cells after immunization with trinitrophenyl-ovalbumin (TNP-Ova), a commonly used antigen for T-cell-dependent antibody responses. Together, our data suggest that the activation of IgD expression by Pten/FoxO1 results in mature B cells that are selectively responsive to multivalent antigen and are capable of initiating rapid GC reactions and T-cell-dependent antibody responses.

**Keywords** B-cell differentiation; FoxO1; immune response; Pten; tolerance
**Subject Categories** Immunology; Signal Transduction
**The EMBO Journal** (2019) 38: e100249

## Introduction

B-cell development is tightly regulated by a network of multiple signaling cascades and a diverse set of transcription factors. Together they control the somatic rearrangement of the *immunoglobulin* (*Ig*) *variable* (*V*), *diversity* (*D*), and *joining* (*J*) gene segments, as well as differentiation, proliferation, and selection events at several B-cell developmental stages. The B-cell antigen receptor (BCR) regulates B-cell development by mediating the selection of functional and self-tolerant B cells, thereby ensuring immune protection and avoiding autoimmunity (Shlomchik, 2008; Pelanda & Torres, 2012).

Phosphoinositide 3-kinase (PI3K) is part of a central BCR-associated signaling axis that regulates diverse key functions during B-cell development. Upon stimulation of the BCR (Harwood & Batista, 2008; Depoil *et al*, 2009), PI3K phosphorylates phosphatidylinositol 4,5-bisphosphate (PIP$_2$) to phosphatidylinositol 3,4,5-trisphosphate (PIP$_3$), thus promoting the recruitment of pleckstrin-homology (PH) domain-containing proteins such as the serine/threonine kinase Akt (also known as PKB), Bruton's tyrosine kinase (Btk), and phospholipase C (PLC)-γ2 to the plasma membrane (Anderson *et al*, 1998; Deane & Fruman, 2004). The PI3K/Akt axis regulates activation or inactivation of multiple cytosolic or nuclear downstream targets including the forkhead box class O (FoxO) transcription factors that are inactivated upon Akt phosphorylation (Yusuf *et al*, 2004; Greer & Brunet, 2005). PI3K controls *Ig heavy chain* (*IgH*) and *light chain* (*IgL*) gene recombination by regulating FoxO1 expression, which is required for the induction of *recombination-activating genes (Rag) 1* and *2* (Amin & Schlissel, 2008; Dengler *et al*, 2008; Herzog *et al*, 2008, 2009). Since FoxO1 activity depends on phosphatase and tensin homolog (Pten), the catalytic PI3K antagonist, which dephosphorylates PIP$_3$ to PIP$_2$ (Maehama & Dixon, 1998; Leslie & Downes,

1  Institute of Immunology, Ulm University Medical Center, Ulm, Germany
2  Department of Molecular Immunology, Biology III, Faculty of Biology, Albert-Ludwigs University of Freiburg, Freiburg, Germany
3  Division of Genetics, Department of Biology, Friedrich Alexander University Erlangen-Nürnberg, Erlangen, Germany
4  Cellular and Molecular Immunology, Georg August University Göttingen, Göttingen, Germany
   *Corresponding author. Tel: +49 731 500 65200; Fax: +49 731 500 65202; E-mail: hassan.jumaa@uni-ulm.de
   †Present address: Clinical Cytomics Facility, University Institute of Clinical Chemistry, Inselspital, Bern University Hospital, Switzerland
   ‡Present address: Lab Hogan, Division of Signaling and Gene Expression, La Jolla Institute for Allergy and Immunology, La Jolla, CA, USA

2002), inactivation of *Pten* or *FoxO1* at an early stage of B-cell development leads to a largely identical block in B-cell development (Dengler *et al*, 2008; Alkhatib *et al*, 2012; Setz *et al*, 2018).

In early B cells, PI3K controls *Ig* gene recombination, and at later stages of development, it regulates the germinal center (GC) reaction in the secondary lymphoid organs where B cells undergo somatic hypermutation (SHM) and class switch recombination (CSR; Victora & Nussenzweig, 2012; Dominguez-Sola *et al*, 2015; Sander *et al*, 2015). These processes, which are a prerequisite for the production of high-affinity antibodies of the IgG, IgA, or IgE isotype, critically depend on activation-induced cytidine deaminase (AID; Stavnezer & Schrader, 2014), which is also regulated by FoxO1 (Omori *et al*, 2006; Dengler *et al*, 2008). Furthermore, FoxO1 has been shown to be essential for the transcription factor network that directs activated B cells into GC reactions by inducing the expression of chemokine receptors that regulate B-cell localization in GCs (Allen *et al*, 2004; Dengler *et al*, 2008; Dominguez-Sola *et al*, 2015; Sander *et al*, 2015; Inoue *et al*, 2017). In line with this finding, Pten-deficient B cells are prone to differentiate into plasma cells that secrete antibodies of the IgM isotype, as shown by highly elevated serum concentrations of IgM, while levels of IgA and IgG are reduced (Suzuki *et al*, 2003; Omori *et al*, 2006). Inhibition of PI3K in Pten-deficient B cells restores the ability of these cells to undergo SHM and CSR by inducing AID (Omori *et al*, 2006).

The majority of mature B cells that circulate between the secondary lymphoid organs belong to the conventional B-2 B-cell compartments such as the follicular (Fo.B) and marginal zone (MZ.B) B cells found in the spleen. Fo.B cells participate in both T-cell-dependent and T-cell-independent immune responses and following affinity maturation and CSR in the GC, B-2 B cells can develop into high-affinity memory isotype-switched B cells or plasma cells (Martin *et al*, 2001; McHeyzer-Williams, 2003). While early developmental stages almost exclusively show IgM BCR expression, mature Fo.B cells exhibit a characteristic upregulation of IgD BCR accompanied by downregulation of IgM. Although this regulation is evolutionary conserved, the role of IgD BCR in B-cell function remains unclear (Chen & Cerutti, 2010, 2011; Ubelhart & Jumaa, 2015; Gutzeit *et al*, 2018). Together with previous data (Kim *et al*, 2006), we recently proposed that IgD is specialized for recognition of antigen encountered in the form of immune complexes (Ubelhart *et al*, 2015) and that this optimizes IgD BCR for T-cell-dependent immune reactions (Roes & Rajewsky, 1993).

In contrast to B-2 B cells, increased amounts of IgM BCR are observed on B-1 B cells which are involved in the earliest antibody responses by secreting mainly autoreactive "natural" IgM antibodies (Murakami *et al*, 1992; Berland & Wortis, 2002). Interestingly, conditional Pten inactivation and the consequently elevated PI3K signaling lead to increased numbers of B-1 B cells (Suzuki *et al*, 2003). In contrast, mice with impaired PI3K signaling due to defective p110δ, the hematopoietic cell-specific catalytic subunit of PI3K, show severely reduced B-1 B-cell numbers (Clayton *et al*, 2002; Jou *et al*, 2002; Okkenhaug *et al*, 2002). Thus, in addition to the important role of PI3K for the generation and maintenance of B cells, it seems to determine the fate of developing B cells (Srinivasan *et al*, 2009; Ramadani *et al*, 2010).

In this study, we demonstrate by a series of *in vitro* and *in vivo* experiments that increased PI3K signaling suppresses IgD expression. Moreover, we show that IgD BCR activation requires polyvalent antigen and is optimized for T-cell-dependent immune responses (Kim *et al*, 2006).

## Results

### Pten is required for receptor editing

Conditional inactivation of Pten or FoxO1 in B cells results in an early developmental block most likely due to the inability of activating *Ig* gene rearrangement (Amin & Schlissel, 2008; Dengler *et al*, 2008; Herzog *et al*, 2008; Alkhatib *et al*, 2012; Setz *et al*, 2018; Fig 1A). To confirm this, we conditionally inactivated *Pten* in *3-83^{ki}* mice that carry knock-in cassettes for *heavy chain* (*HC*) and *light chain* (*LC*) of an autoreactive BCR that recognizes the MHC class I allele H2-K$^b$ with high affinity as compared to the H2-K$^d$ allele (Fig 1A). For unknown reasons, in this model receptor editing by secondary *IgL* gene rearrangement can also be observed on the H2-K$^d$ background leading to loss of the knock-in *LC* in the *3-83^{ki}* mice (Pelanda *et al*, 1997; Fig 1B and C). By using mb1-cre to conditionally inactivate *Pten* in early B cells expressing the 3-83 BCR, we found that these *3-83 HC* and *LC* knock-ins rescued the block of early B-cell development observed in Pten-deficient B cells in bone marrow and spleen (Fig 1A). On the non-autoreactive H2-K$^d$ background, the Pten-deficient B cells expressed the 3-83 BCR on their surface as measured by staining with the anti-idiotype antibody 54.1 (Fig 1B). However, on the autoreactive H2-K$^b$ background, no BCR was detected on the cell surface (Fig 1A and B). However, neither H2-K$^d$ nor H2-K$^b$ background showed receptor editing as the *3-83 LC* knock-in was readily detected in the genomic DNA of splenic B cells of either background (Fig 1C).

Together, these data suggest that Pten-deficient B cells cannot edit an autoreactive BCR specificity (Halverson *et al*, 2004; Lang *et al*, 2016).

### Pten-deficient B cells are capable of acquiring an anergic phenotype

Our results show that Pten-deficient *3-83^{ki}* B cells lack surface BCR expression on the H2-K$^b$ background despite the defect in receptor editing. To confirm the expression of the knock-in BCR components, we performed intracellular IgM staining and found that almost all Pten-deficient *3-83^{ki}* B cells show IgM expression in bone marrow and spleen, while Pten-deficient B cells lacking the knock-in cassettes showed only a minor fraction of IgM-expressing cells (Fig 2A).

These data suggest that Pten-deficient *3-83^{ki}* B cells internalize or downregulate surface BCR on the autoreactive H2-K$^b$ background. Since downregulation of the autoreactive BCR renders B cells unresponsive and is known as B-cell anergy, these data suggest that Pten is not required for anergy induction. To test this directly, we treated splenic B cells from Pten-deficient *3-83^{ki}* mice of the different H2-K backgrounds with anti-BCR (anti-κLC) antibodies and monitored the subsequent (calcium) Ca$^{2+}$ response (Fig 2B and Appendix Fig S1A). We found that, in contrast to the H2-K$^d$ background, Pten-deficient *3-83^{ki}* B cells on the autoreactive H2-K$^b$ background failed to mount a Ca$^{2+}$ response, which is in full agreement with their lack of surface BCR on this background (Fig 2B). Moreover, serum

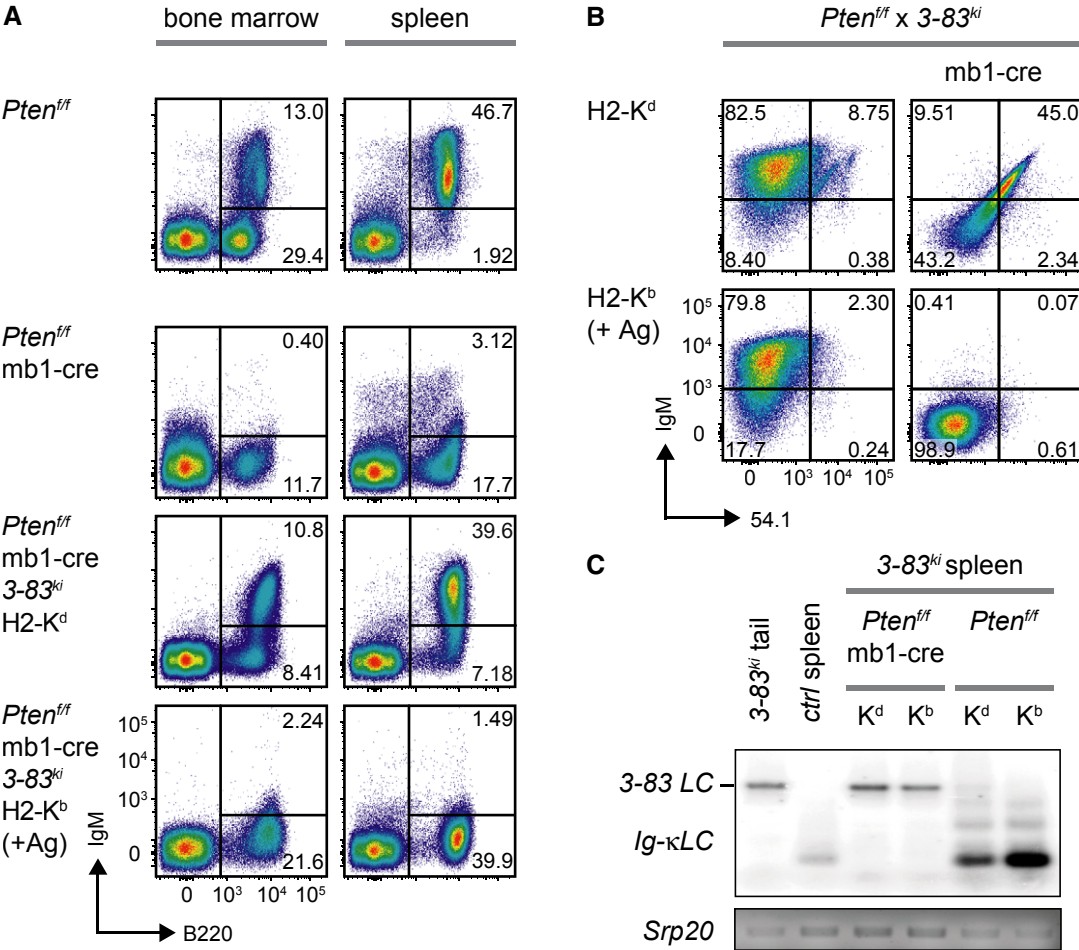

**Figure 1. Pten is required for receptor editing.**

A Representative analysis of B220 and IgM surface expression in bone marrow (left) and spleen cells (right) of mice from the indicated genotypes.

B Representative flow cytometric analysis of splenocytes from mice of the indicated genotypes (pre-gated on B cells: B220$^+$/CD19$^+$) for surface expression of IgM and the 3-83 idiotype (54.1). Shown data are representative of 11–35 individual mice per genotype.

C PCR fragments amplified with specific primers for $V\kappa$ and $J\kappa5$ from genomic DNA of purified splenic B cells from *Pten$^{f/f}$* × *3-83$^{ki}$* and *Pten$^{f/f}$* × mb1-cre × *3-83$^{ki}$* mice on the respective backgrounds. Genomic tail DNA from a *3-83$^{ki}$* mouse and DNA from purified splenic B cells of a control (*ctrl*) mouse were used as controls. PCR for the splicing factor *Srp20* served as a loading control. K$^b$ and K$^d$ indicate the respective background of the mice (H2-K$^b$: +Ag).

antibody concentrations were markedly reduced in *3-83$^{ki}$* mice on the autoreactive H2-K$^b$ background (Fig 2C) although the total number of B cells appeared to be increased compared to Pten-deficient mice lacking the *3-83* knock-in cassettes (Fig 2D and E). Interestingly, anergic Pten-deficient *3-83$^{ki}$* B cells on the autoreactive H2-K$^b$ background did not show a shortened life span upon *in vitro* culture as compared to their counterparts on the non-autoreactive H2-K$^d$ background (Appendix Fig S1B).

These data indicate that Pten-deficient B cells expressing an autoreactive BCR downregulate surface BCR expression and become unresponsive to stimulation by anti-BCR antibodies suggesting that induction of anergy does not require Pten.

## IgD BCR expression requires Pten

Surprisingly, we found that Pten-deficient *3-83$^{ki}$* B cells were unable to upregulate IgD BCR expression (Fig 3A) despite the

normal IgM BCR expression on the H2-K$^d$ background (Figs 1A and 2A). To investigate whether Pten is required for the activation of IgD expression independent of the BCR specificity, we conditionally inactivated Pten in B cells from *MD4*-transgenic (*MD4$^{tg}$*) mice expressing transgenes for *HC* and *LC* of a hen-egg lysozyme (HEL)-specific BCR, which allow the generation of IgM and IgD BCR. In the presence of Pten and the cognate antigen HEL (*MD4* × *ML5* double-transgenic mice), IgM is largely downregulated, while IgD surface expression is maintained (Goodnow *et al*, 1988).

In agreement with the proposed role of Pten for IgD expression, *MD4$^{tg}$* B cells showed significant reduction of IgD expression in the absence of Pten as compared with control B cells (Fig 3B and C). Introducing the antigen by crossing in the *ML5*-transgenic (*ML5$^{tg}$*) mice failed to upregulate IgD expression (Fig 3B). Interestingly, the amounts of serum IgM were similar in *MD4* single- and *MD4* × *ML5* double-transgenic mice

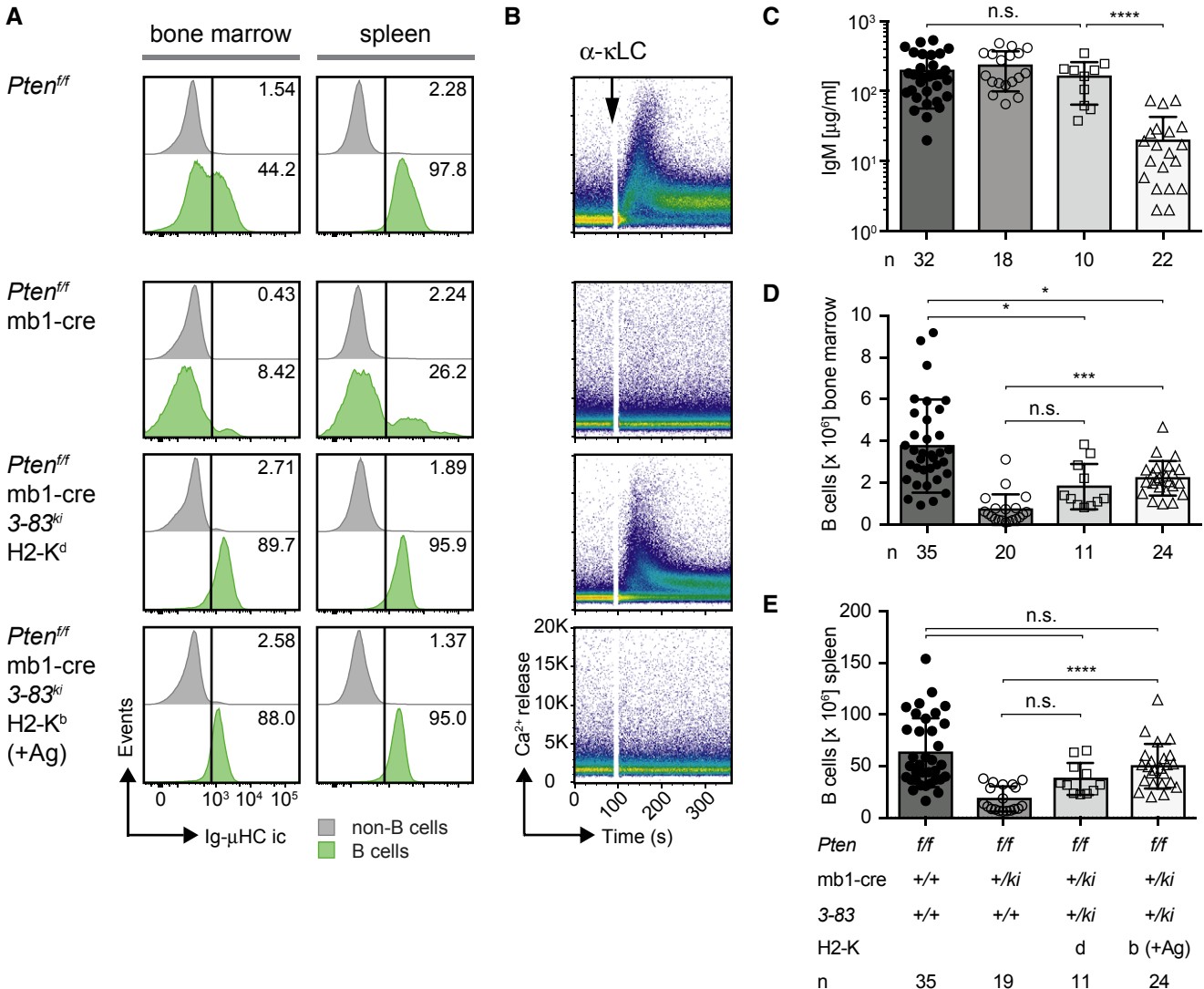

**Figure 2. Pten-deficient B cells are capable of acquiring an anergic phenotype.**

A    Intracellular expression of IgM (Ig-μHC ic) in bone marrow (left) and spleen cells (right) was determined by flow cytometry and compared between the populations of B cells (green, identified by B220 and CD19 expression) and non-B cells (gray). Numbers in the histograms indicate the percentages of the positive populations. Figures are representative of 11–35 individual mice per genotype.

B    Intracellular Ca²⁺ influx was measured in CD90.2/Thy1.2⁻ splenocytes derived from mice of the indicated genotypes following stimulation with 10 μg/ml α-κLC antibody. Figures are representative of at least three individual mice per genotype.

C    Serum IgM concentrations measured in mice of the indicated genotypes. Mean ± SD, symbols represent IgM concentrations from individual mice ($n$). Statistical significance was calculated by using the Kruskal–Wallis test (see also Appendix Table S1), n.s. = not significant; ****$P \leq 0.0001$.

D, E  Absolute numbers of B cells in bone marrow (D) and spleens (E) from mice of the respective genotypes, determined by B220/CD19 surface expression. Mean ± SD, symbols indicate the numbers of B cells in individual mice ($n$). Statistical significance was calculated by using the Kruskal–Wallis test (see also Appendix Tables S2 and S3), n.s. = not significant; *$P \leq 0.05$; ***$P \leq 0.001$; ****$P \leq 0.0001$.

(Fig EV1A), while the levels of HEL-specific antibody were reduced in sera from *MD4 × ML5* double-transgenic mice (Fig 3D).

Moreover, we found abnormal expression of CD23 in Pten-deficient cells (Fig EV1B). To determine whether the development of mature Fo.B cells is disturbed in general or whether Pten deficiency affects only the regulation of IgD and CD23 expression, we analyzed whether follicular structures are present in the spleens of the respective mice (Fig 3E and Appendix Fig S2). In contrast to spleens from control mice, spleens from *Pten*^f/f × mb1-cre mice showed no organized structures for B cells. Introduction of pre-rearranged *Ig* genes restored the presence of IgM-expressing cells and increased the numbers of splenic B cells in the follicles. Still, the areas occupied by follicular B cells between the marginal and T-cell zone were smaller in the absence of Pten as compared with controls. This suggests that the development of Fo.B cells is affected in the absence of Pten/FoxO1 function in addition to the impaired expression of Fo.B cell markers such as IgD and CD23.

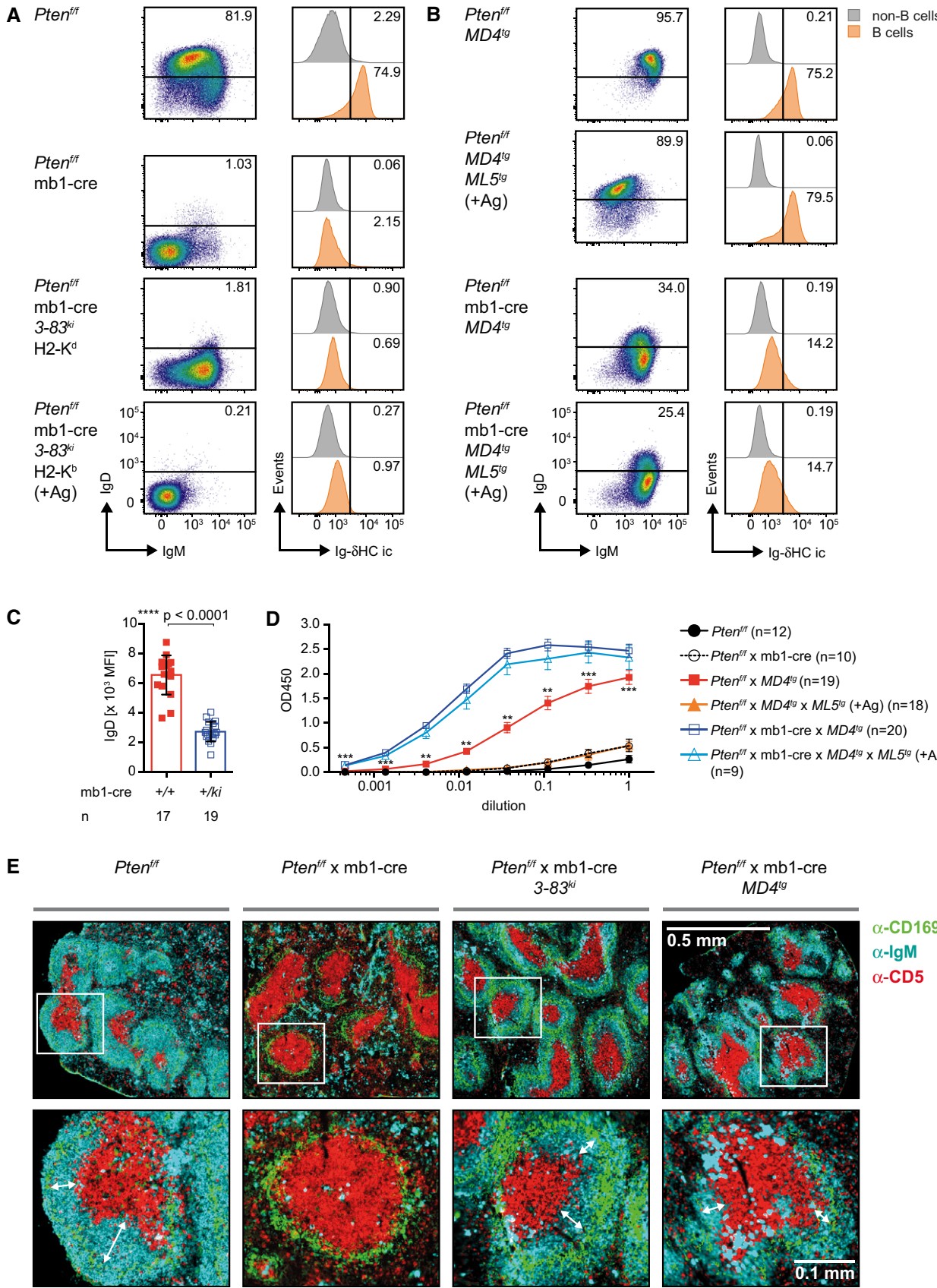

Figure 3.

**Figure 3.  IgD BCR expression requires Pten.**

A, B   Splenocytes from mice of the indicated genotypes, (A) $Pten^{f/f}$, $Pten^{f/f} \times$ mb1-cre, *and* $Pten^{f/f} \times$ mb1-cre 3-83$^{ki}$ mice on the respective backgrounds, (B) $Pten^{f/f} \times MD4^{tg}$ *and* $Pten^{f/f} \times$ mb1-cre $\times MD4^{tg}$ mice in presence and absence of antigen ($ML5^{tg}$), were analyzed by flow cytometry. B cells were pre-gated by B220 and CD19 expression, and the IgM/IgD surface expression (dot plots, left) was determined. Histograms (right) show a comparison of intracellular IgD expression (Ig-δHC ic) between the splenic populations of B cells (orange, identified by B220 and CD19 expression) and non-B cells (gray). Numbers in the histograms indicate the percentages of the positive populations. Data are representative of 11–35 individual mice per genotype.

C   Quantification of surface IgD mean fluorescence intensity (MFI) in $Pten^{f/f} \times$ MD4-transgenic mice in the presence (mb1-cre +/+) and absence (mb1-cre +/ki) of Pten. Single dots represent the average MFI of four independent measurements per mouse (*n*), mean ± SD. Statistical significance was calculated by using an unpaired two-tailed *t*-test.

D   Serum anti-HEL IgM titers. Sera from $MD4^{tg}$ mice of the indicated genotypes were adjusted to an IgM concentration of 500 μg/ml and applied in triplicates to HEL-coated plates in dilution steps of 1:3, mean ± SEM. Statistical significance was calculated by using the Kruskal–Wallis test (see also Appendix Table S4). Statements of significance in the figure refer to the comparison between $Pten^{f/f} \times MD4^{tg}$ (filled red squares) and $Pten^{f/f} \times MD4^{tg} \times MD5^{tg}$ sera (filled orange triangles), **$P \leq 0.01$; ***$P \leq 0.001$.

E   Immunohistochemistry of sections from spleens of $Pten^{f/f}$, $Pten^{f/} \times$ mb1-cre, $Pten^{f/f} \times$ mb1-cre $\times$ 3-83$^{ki}$, and $Pten^{f/f} \times$ mb1-cre $\times MD4^{tg}$ mice for CD169 (green), CD5 (red), and IgM (cyan) at 10× magnification. Pictures in the second row show enlarged areas indicated by the white squares, respectively. Double arrows indicate the distance between T cell (red) and marginal zone (green). Shown pictures are representative of 2–3 mice per genotype (see also Appendix Fig S2).

## Pten activates IgD expression and development of Fo.B cells

The previous results suggest that Pten is required for efficient generation of Fo.B cells. Therefore, we tested whether Pten downstream elements such as the transcription factor FoxO1 can restore the developmental block observed in Pten-deficient splenic B cells. To this end, we transduced Pten-deficient splenic B cells from $MD4^{tg}$ single-transgenic mice with a constitutively active form of FoxO1, FoxO1-A3 (Alkhatib *et al*, 2012), and monitored IgD and CD23 expression (Fig 4A). We found that Pten-deficient $MD4^{tg}$ B cells transduced with FoxO1-A3 show upregulation of IgD and CD23 expression (Fig 4B).

In addition, we purified Pten-deficient splenic IgM$^{hi}$/IgD$^-$ B cells expressing endogenous BCR and transduced them with FoxO1-A3 (Fig EV2A and B). The results show that FoxO1-A3 led to increased IgD and CD23 expression on the surface of the transduced splenic B cells (Fig EV2C and D). To test whether this increase was the result of transcriptional activation, we analyzed *Ig-δHC* transcripts (*Ighd*) in the transduced B cells. We found that FoxO1-A3-transduced cells show significantly increased amounts of *Ig-δHC* transcripts as compared with cells transduced with the empty vector (EV; Fig EV2E). In line with this finding, deletion of *FoxO1* in splenic B cells derived from $FoxO1^{f/f}$ mice by introducing a cre-encoding expression vector (Figs 4D and EV2F) resulted in reduced IgD and CD23 surface expression (Fig 4E). As FoxO1 deletion also seemed to affect IgM surface expression (Fig EV2G), we analyzed mRNA levels of *Ig-μHC* (*Ighm*) and *Ig-β* (*Cd79b*). In line with previous studies (Dengler *et al*, 2008), we detected by tendency reduced *Ig-β* expression in the absence of FoxO1, which may also account for the reduced surface IgM expression, as *Ig-μHC* transcripts were not downregulated (Fig EV2H). The slight changes observed in CD23 expression upon overexpression/deletion of FoxO1 were further confirmed by analyzing the transcripts of *Fcer2*, the gene encoding CD23 (Fig 4B and E).

We further examined the expression of the poly-adenylation factors *Cstf64* and *Ell2,* which have been proposed to regulate the expression of the membrane-bound and secreted form of IgM (Takagaki *et al*, 1996; Martincic *et al*, 2009). In addition, we investigated expression of the zinc-finger protein *Zfp318,* which has been reported to regulate *Ig-μ/δHC* pre-mRNA splicing towards the expression of IgD (Enders *et al*, 2014; Pioli *et al*, 2014). Upon ectopic overexpression of FoxO1-A3 in Pten-deficient IgM$^{hi}$/IgD$^-$ splenic B cells (as shown in Fig 4A and B), we detected reduced transcript levels of *Ell2* and elevated mRNA expression of *Zfp318* (Fig 4C). Notably, deletion of *FoxO1* (as shown in Fig 4D and E) led to increased *Ell2* transcript levels, whereas expression of *Zfp318* was significantly reduced (Fig 4F). However, transcript levels of *Cstf64* were not significantly changed in both experimental setups (Fig 4C and F). This reciprocal regulation of *Ell2* and *Zfp318* following constitutive activation or deletion of FoxO1, respectively, suggests that FoxO1 controls IgD expression via regulating transcription of *Ell2* and *Zfp318*. To test whether FoxO1 directly binds to regulatory elements in the gene loci of *Zfp318*, *Ell2,* or *Fcer2*, we analyzed available data on genome-wide FoxO1 occupancy in B cells.

Analysis of ChIP-Seq data, chromatin immunoprecipitation combined with deep DNA sequencing, revealed no FoxO1-binding within the genes of interest (Lin *et al*, 2010). To further confirm these results, we designed a ChIP assay on sorted mature splenic wild-type (WT) B cells (Appendix Fig S3). In agreement with the published results, our ChIP assay using mature splenic WT B cells showed no direct FoxO1-binding to *Zfp318*, *Ell2,* or *Fcer2* gene loci (Appendix Fig S3; Appendix Tables S12–S14).

Together, these data suggest that a Pten-regulated program controls FoxO1 activity and activates the expression of IgD and CD23 by an indirect mechanism, thereby enabling the development of mature Fo.B cells.

## Monovalent antigen controls IgD BCR activation

Together with previous data (Ubelhart *et al*, 2015), our results suggest that Pten controls the responsiveness of B cells by regulating the IgD-to-IgM ratio.

To confirm that this central role of IgD is independent of the cell line and its origin, we used the human Burkitt lymphoma B-cell line Ramos and knocked out the *HC* and *LC* genes by the CRISPR/Cas technique (He *et al*, 2018; Fig EV3A and B). The resulting HC and LC knockout (HL-KO) Ramos cells were reconstituted with retroviral expression vectors (Fig 5A) encoding murine anti-HEL IgM (HH10-mu IgM) or IgD (HH10-mu IgD). Both BCR isotypes were similarly expressed (Fig 5B) and showed comparable HEL-binding (Fig 5C). Consistent with our previous results, HH10-mu IgD remained

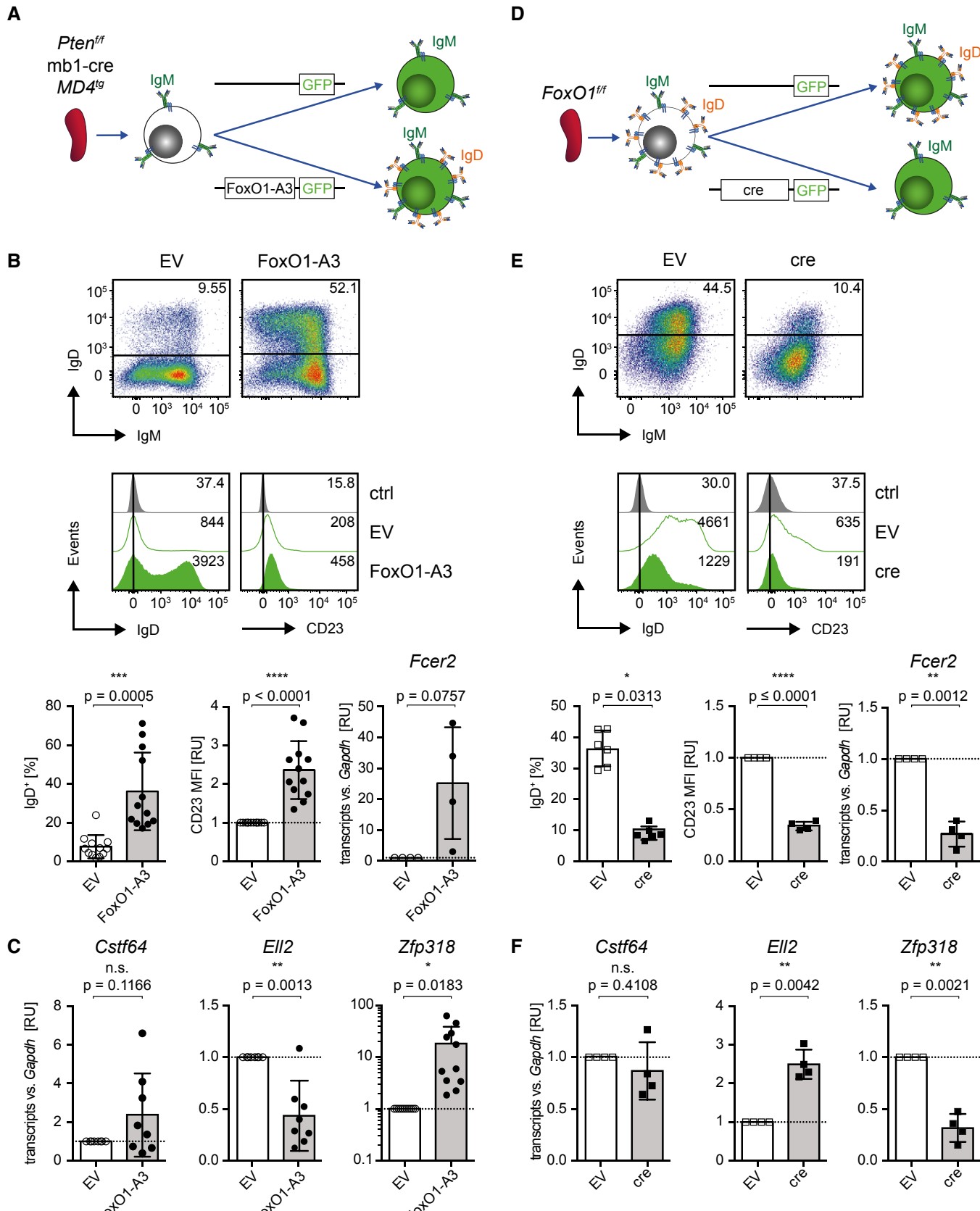

**Figure 4.**

◄

**Figure 4. Pten activates IgD expression and development of Fo.B cells.**

A   Schematic overview of the transduction procedure: Purified *Pten*^f/f^ × mb1-cre × *MD4*^tg^-derived mature splenic B cells were cultured in the presence of LPS 2.5 μg/ml for 1.5 days and subjected to retroviral transduction with expression vectors encoding the constitutively active FoxO1 (FoxO1-A3) or the empty vector (EV), respectively.

B   Three days posttransduction, the surface expression of IgM/IgD (dot plots, top), IgD, and CD23 (histograms, bottom) was measured by flow cytometry and compared between EV- and FoxO1-A3-transduced cells. Fluorescence minus one (FMO) stainings, lacking α-IgD or α-CD23 antibodies, respectively, served as controls (ctrl). Bar diagrams below flow cytometric data display the percentages of IgD^+ cells (*n* = 12; left) and MFI of CD23 (as relative units, RU), normalized to the MFI measured in the respective EV-transduced cells (*n* = 12; middle). *Fcer2* transcript levels were measured by quantitative reverse transcriptase (qRT)–PCR (right; *n* = 4). Mean ± SD, symbols indicate data from individual mice. Statistical significance of IgD expression was determined by applying the Wilcoxon matched-pairs signed rank test for CD23 and by applying the one-sample two-tailed *t*-test for *Fcer2* expression.

C   *Cstf64* (*n* = 8), *Ell2* (*n* = 8), and *Zfp318* (*n* = 11) expression levels were measured at 3 days posttransduction by qRT–PCR. FoxO1-A3- and EV-transduced splenic B cells from Fig 4B were FACS-purified according to the gating strategy shown in Fig EV2B. Mean ± SD, symbols indicate the expression in individual mice. Statistical significance was calculated by using the one-sample two-tailed *t*-test.

D   Schematic overview of transduction procedure: Purified *FoxO1*^f/f^-derived mature splenic B cells were cultured in the presence of 2.5 μg/ml LPS for 1.5 days and subjected to retroviral transduction with expression vectors encoding either cre or EV, respectively.

E, F   Identical to Fig 4B and C using cells described in Fig 4D, E: IgD (*n* = 6), CD23 (*n* = 4), and *Fcer2* (*n* = 4). F: *Cstf64* (*n* = 4), *Ell2* (*n* = 4), and *Zfp318* (*n* = 4).

unresponsive to monovalent soluble HEL (sHEL) in contrast to complex multivalent HEL (cHEL), whereas HH10-mu IgM was equally responsive to both sHEL and cHEL (Fig 5D). Furthermore, as observed in our previous experimental system (Ubelhart *et al*, 2015), sHEL interfered with cHEL-induced signaling of HH10-mu IgD, while HH10-mu IgM signaling remained unaffected (Fig 5E). Thus, the ratio of monovalent to multivalent antigen determines the threshold of IgD activation.

Furthermore, we assessed whether human IgM and IgD isotypes recapitulate the differential responsiveness toward monovalent antigen similar to their murine BCR counterparts. We therefore replaced the constant regions of the HCs by the matching parts of human origin and generated human IgM (HH10-hu IgM) or IgD (HH10-hu IgD) BCRs (Fig 5F), respectively. Indeed, the HL-KO cells expressing HH10-hu IgD required cHEL for Ca^{2+} mobilization, whereas HH10-hu IgM-expressing cells were equally responsive to both sHEL and cHEL (Fig 5G–I). Moreover, the interference of sHEL with cHEL-induced HH10-hu IgD signaling remained conserved (Fig 5J).

Taken together, these data demonstrate that differential responsiveness of IgM and IgD toward monovalent antigen is an inherent isotype-specific feature that is independent of cell lines, inducible signaling machinery or BCR constant regions from either mouse or human.

## IgD expression modulates B-cell responsiveness

Since B-cell responsiveness is determined both by the ratio of IgM to IgD surface expression and by the ratio of mono- to multivalent antigen, we tested whether the defective upregulation of IgD in the *MD4* × *ML5* double-transgenic B cells in absence of Pten was accompanied by an increased response of the corresponding B cells to sHEL as compared with multivalent cHEL. Comparison of surface BCR expression revealed that, in absence of Pten, both MD4 single- and *MD4* × *ML5* double-transgenic B cells show equally high IgM expression but strongly reduced IgD expression as compared with their Pten-sufficient counterparts (Fig 6A). As expected, B cells expressing an increased amount of IgM BCR showed strong Ca^{2+} influx in response to sHEL treatment while B cells expressing IgD BCR responded only to cHEL (Fig 6B and C). This is in agreement with the proposed essential role of Pten in the activation of IgD expression and with the selective responsiveness of IgD toward multivalent antigen.

Interestingly, Pten inactivation in B cells results in increased compartments of MZ.B and CD21^lo B cells (Suzuki *et al*, 2003; Setz *et al*, 2018), which show lower IgD expression as compared with Fo.B. Although splenic Fo.B and MZ.B cells also differ in further properties such as CD21 and CD23 expression, MZ.B cells induced a stronger Ca^{2+} influx upon stimulation (Fig EV3C–E).

Together these data suggest that Pten modulates B-cell responsiveness and B-cell development by activating IgD expression and thereby altering the IgD-to-IgM ratio.

## Abnormal germinal center reaction in IgD-deficient mice

Our findings suggest that Pten-mediated FoxO1 activation is required for peripheral B-cell development and particularly for the generation of mature Fo.B cells. Recent data showed that FoxO1 is required for the GC reaction, which plays a central role in immune responses and the generation of high-affinity antibodies (Dominguez-Sola *et al*, 2015; Sander *et al*, 2015). Together with our finding that FoxO1 activates IgD expression and with previous studies reporting defective early antibody responses in IgD-deficient mice (Roes & Rajewsky, 1993), these data imply that activation of IgD expression may be an important step in the FoxO1-mediated GC reaction.

To test whether proper IgD expression is an important part of the FoxO1-mediated regulation of the GC reaction, we compared the GC formation in IgD-deficient mice with that of WT mice after immunization with 2,4,6-trinitrophenyl hapten conjugated to ovalbumin (TNP-Ova), a commonly used multivalent antigen for T-cell-dependent antibody responses (Fig 7A). We monitored the generation of GC B cells (B220^+/CD38^−/CD95^+/GL7^+) at days 0, 4, 7, 10, and 14 following immunization by flow cytometry (Fig 7B and C; Dominguez-Sola *et al*, 2012; Inoue *et al*, 2017). We found that IgD-deficient mice exhibit a delay in GC reaction as shown by their markedly reduced GC cells at days 4 and 7 of immunization (Fig 7C). To confirm these data, we analyzed spleen sections from the immunized mice at days 4, 7, and 10 postimmunization. In full agreement, we found that, in contrast to WT controls, GCs can hardly be detected in IgD-deficient mice at days 4 and 7 of immunization (Fig 7D and E). On day 10 after immunization, the GC cells in IgD-deficient mice were mostly localized in multiple small aggregations as compared to the compact GC structures observed in WT mice (Fig 7E and F; Sander *et al*, 2015; Degn *et al*, 2017; Inoue *et al*, 2017). Moreover,

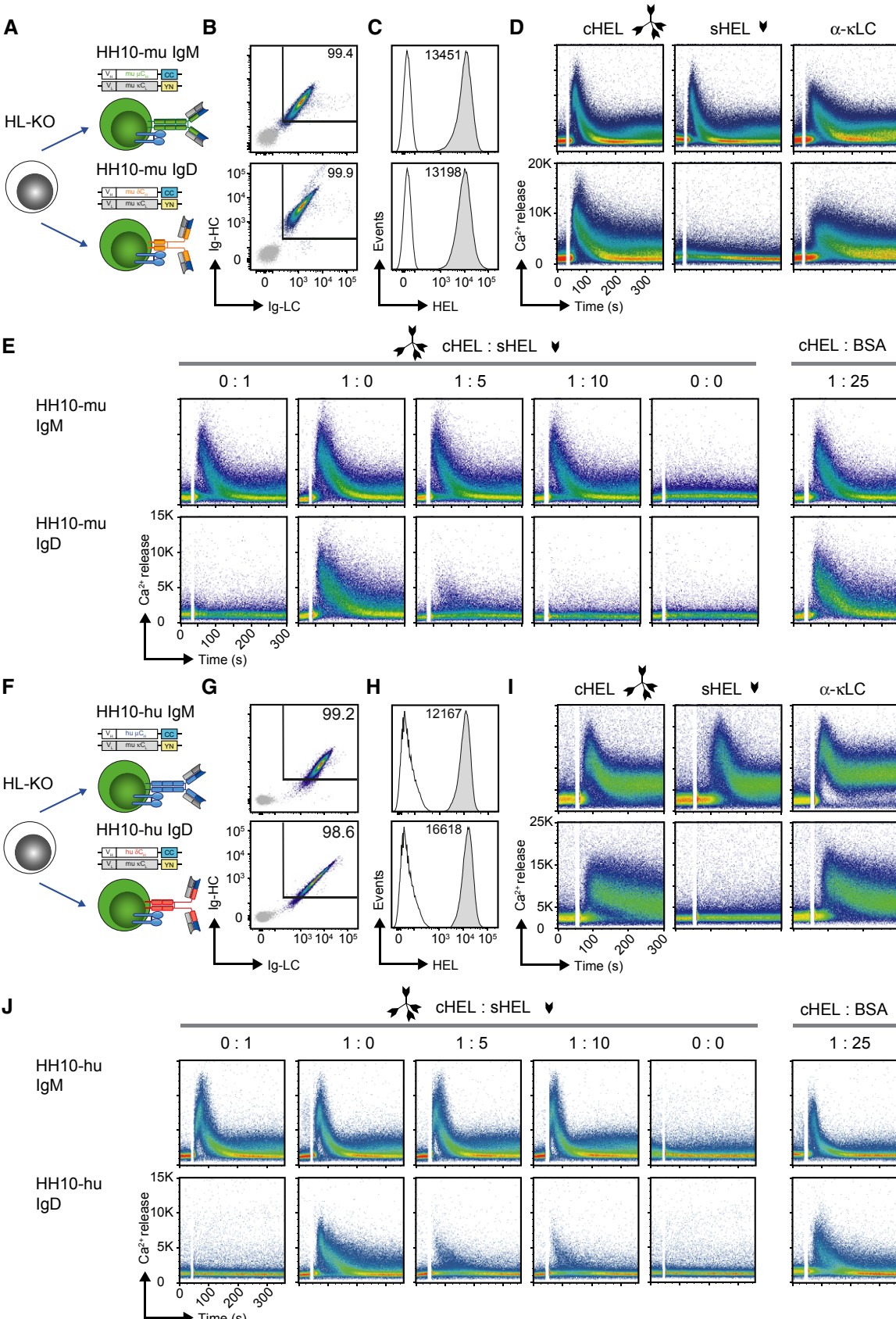

**Figure 5.**

**Figure 5.  Monovalent antigen controls IgD BCR activation.**

A    Schematic overview of reconstitution of BCR-deficient Ramos (HL-KO) cells with murine HEL-specific BCR of IgM (HH10-mu IgM) or IgD (HH10-mu IgD) isotype, respectively.

B    Representative flow cytometric analysis of Ramos cells reconstituted with HH10-mu IgM or HH10-mu IgD, respectively. Cells were stained with α-murine μHC, δHC, and κLC antibodies, respectively. EV-transduced HL-KO cells, expressing GFP only, were used as negative control (gray).

C    Representative flow cytometric analysis of HEL-binding in reconstituted Ramos cells after staining with fluorescently labeled HEL. EV-transduced HL-KO cells, expressing GFP only, were used as negative control (gray).

D    Representative intracellular Ca$^{2+}$ influx in HH10-mu IgM- and HH10-mu IgD-expressing cells upon stimulation with multivalent HEL (complex cHEL), monovalent (soluble sHEL; both at a concentration of 1 μg/ml), or 10 μg/ml α-mouse κLC antibody, respectively.

E    Representative intracellular Ca$^{2+}$ influx of HH10-mu IgM- (top) and HH10-mu IgD- (bottom) expressing cells upon stimulation with indicated ratios of 1 μg/ml cHEL and sHEL, and a 1:25 mixture of cHEL with bovine serum albumin (BSA), where 1 = 1 μg/ml.

F    Schematic overview of reconstitution of BCR-deficient Ramos (HL-KO) cells with HH10-specific human IgM (HH10-hu IgM) or IgD (HH10-hu IgD) BCR isotype, respectively.

G–J  Identical to Fig 5B–E using HH10-hu IgM- and HH10-hu IgD-expressing cells instead of HH10-mu IgM- and HH10-mu IgD-expressing cells, respectively. Data shown in B–E and G–J are representative of at least three independent experiments.

sera from TNP-Ova-immunized IgD-deficient mice contained less TNP-specific IgG compared to sera from immunized WT mice (Fig 7G).

Immunization with sheep red blood cells (SRBC), a very strong immunogen bearing an unlimited number of different antigens, which is thought to activate B cells by pattern recognition receptors (PRRs; Loetsch *et al*, 2017), led to similar amounts of GC B cells in both WT and IgD$^{-/-}$ mice (Fig EV4). Thus, similar to earlier results (Roes & Rajewsky, 1993), our data suggest that efficient immune responses against some pathogens require IgD for directing B cells into GC reactions.

## Discussion

The data presented in this study show that Pten activates IgD expression via FoxO1 and that IgD-expressing B cells are selectively responsive to multivalent antigen. Moreover, we report that IgD-deficient mice show a delayed GC reaction to TNP-Ova, as compared to wild-type mice, suggesting that IgD expression allows efficient recruitment of B cells into germinal centers.

The data presented in this study confirm previous findings (Roes & Rajewsky, 1993; Kim *et al*, 2006; Ubelhart *et al*, 2015) and show that IgD requires multivalent antigen for activation. Furthermore, the fact that the ratio of IgD to IgM is regulated during B-cell development suggests that B-cell responsiveness undergoes a shift toward recognition of immune complexes by increasing the amount of IgD during development. By using the human mature Burkitt lymphoma B-cell line Ramos, expressing endogenous signaling machinery, we demonstrate that the requirement of multivalent antigen for IgD activation is intrinsic, as neither murine pro-B cells (Ubelhart *et al*, 2015) nor human mature Ramos cells show a Ca$^{2+}$ influx when treated with monovalent antigen. Moreover, our experiments show that the responsiveness of IgD to polyvalent antigen is inhibited by increasing amounts of monovalent antigen suggesting a regulatory role for IgD, but not for IgM BCR. An important consequence of this differential responsiveness is a dynamic scenario for B-cell activation, which is determined by the ratio of both IgM to IgD and monovalent to multivalent antigen. Thus, mature B cells characterized by increased IgD expression require polyvalent antigens or antigen complexes for their efficient stimulation while interference from monovalent antigen elevates the activation threshold for antigen complexes. This regulation ensures that mature B-cell activation and subsequent immune responses including the production of

highly specific antibodies are mounted on earlier infection phases during which immune complexes are formed and delivered by the complement system and natural IgM. Activation of mature B cells by such antigen-containing immune complexes, either alone or presented via Fc-receptors on other cells, not only integrates B-cell immune responses into a network of ongoing immune reaction but might also prevent unwanted activation by monomeric foreign or self-antigen. Thereby, the reduced amounts of IgM BCR on mature B cells might still be involved in the constant stimulation that has been described for mature B cells (Zikherman *et al*, 2012).

Thus, B-cell maturation results in cells that rapidly provide efficient antibody responses during infections, while ignoring harmless antigen. This selective responsiveness is a hallmark of B-cell maturation and is mediated by IgD BCR, which in contrast to IgM has an extended hinge region that allows efficient binding of immune complexes (Nezlin, 1990). Also, the selective responsiveness of IgD is most likely the reason why B cells from *MD4 × ML5* double-transgenic mice, which express increased amounts of anti-HEL IgD as compared with IgM, fail to respond when stimulated with soluble HEL (Ubelhart *et al*, 2015). Referring to these cells as "unresponsive" or "anergic" is misleading because these cells are responsive to treatment with multimeric HEL complexes or with anti-BCR antibodies (Goodnow *et al*, 1988, 1991; Ubelhart *et al*, 2015). Thus, the proposed anergic state of B cells from *MD4 × ML5* double-transgenic mice is indistinguishable from the mature state of normal B cells (Zikherman *et al*, 2012). Moreover, mutations hampering IgD expression shift the IgD-to-IgM ratio and are expected to alter the responsiveness of B cells.

In fact, decreasing IgD expression in these mice by inactivation of Pten results in IgM being the predominant BCR isotype expressed by *MD4 × ML5* double-transgenic B cells that readily respond to soluble antigen (Browne *et al*, 2009). This suggests that *Pten* inactivation changes the IgD-to-IgM ratio and that the gain in responsiveness of Pten-deficient B cells toward soluble antigen is caused by a developmental block affecting IgD expression. This in addition to the fact that Pten-deficient *3-83$^{ki}$* B cells downregulate IgM BCR expression and do not respond to further stimulation argues against an essential role of Pten in anergy. It is not clear whether Pten deficiency may affect the reported anergy in other transgenic systems testing autoreactive BCRs. However, it should be considered that these systems usually utilize classical transgenes interfering with normal IgD expression (Cambier *et al*, 2007).

Together, our data call for a clear discrimination between mature B cells, which express IgD and are responsive to polymeric

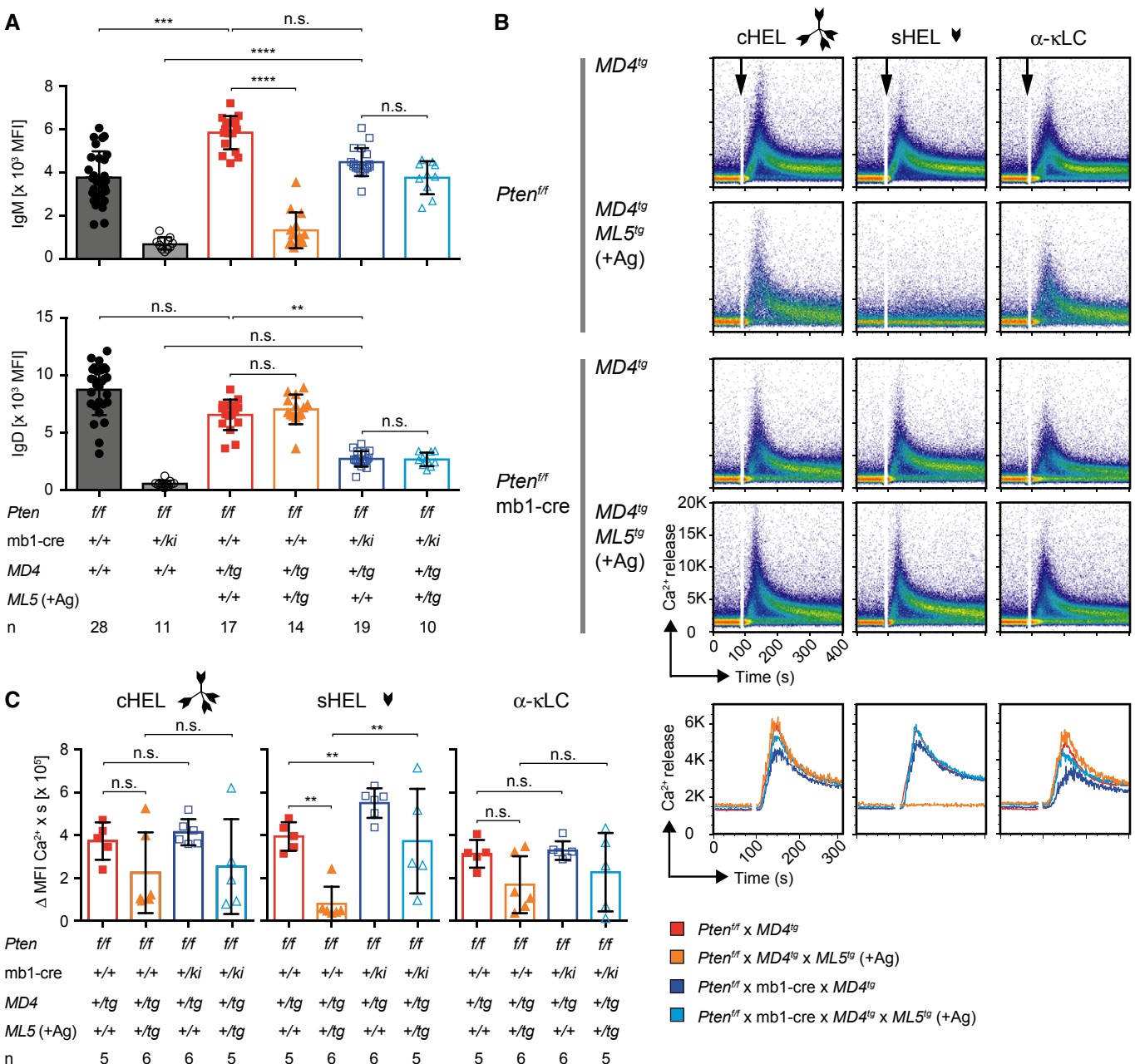

**Figure 6. IgD expression modulates B-cell responsiveness.**

A  Quantification of surface IgM (top) and IgD (bottom) MFI in B cells from mice of the indicated genotypes. Single dots represent the average MFI of four independent measurements per mouse (*n*), mean ± SD. IgD expression of *Pten*[f/f] × *MD4*[tg] and *Pten*[f/f] × mb1-cre × *MD4*[tg] B cells already shown in Fig 3C. Statistical significance was calculated by using the Kruskal–Wallis test (see also Appendix Table S5), n.s. = not significant; **$P \le 0.01$; ***$P \le 0.001$; ****$P \le 0.0001$.

B  Representative intracellular $Ca^{2+}$ influx measured in Thy1.2⁻ splenocytes, derived from mice of the indicated genotypes, upon stimulation with cHEL, monovalent sHEL (both at a concentration of 1 μg/ml) or 10 μg/ml α-mouse κLC antibody, respectively. Line diagrams below the dot plots show the overlaid MFI of the $Ca^{2+}$ influx kinetics displayed in the plots above.

C  Quantification of intracellular $Ca^{2+}$ influx from Fig 6B. The median area under the curve of the $Ca^{2+}$ influx kinetics was calculated, and single dots represent data from individual mice, mean ± SD. Statistical significance was calculated by applying the Mann–Whitney *U*-test (see also Appendix Table S6), n.s. = not significant, **$P \le 0.01$.

antigen or anti-BCR stimulation, and anergic cells that downregulate BCR expression and become unresponsive to BCR stimulation in general. According to this discrimination, *MD4 × ML5* B cells are not anergic and most likely correspond to mature B cells that acquire a selective responsiveness by increased IgD expression. Instead, B cells that downregulate BCR expression of both IgM and IgD represent the genuine anergic cells as they cannot be triggered by their BCR.

Corinna S Setz et al

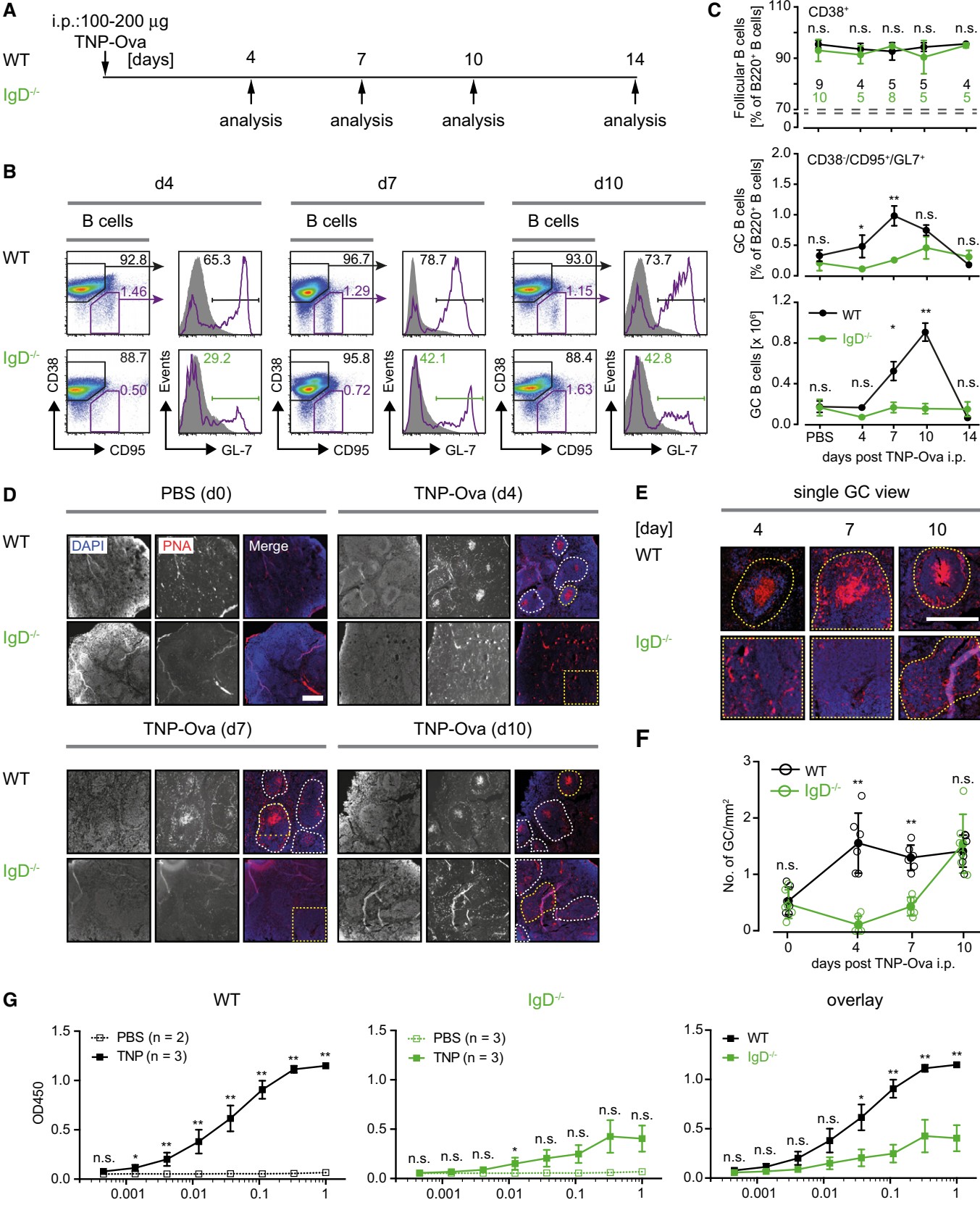

Figure 7.

**Figure 7. Abnormal germinal center reaction in IgD-deficient mice.**

A Schematic overview of the procedure of mouse immunization with 2,4,6-trinitrophenyl hapten conjugated to ovalbumin (TNP-Ova) and analysis of germinal center (GC) reactions at different time intervals. Control animals were injected intraperitoneally (i.p.) with PBS and quantified as day 0 of immunization.

B Analysis of GC B cells in splenic B220$^+$ B cells from wild-type (WT, top) and IgD$^{-/-}$ (bottom) animals after 4, 7, and 10 days (left to right) postimmunization. Percentages of follicular B cells (Fo.B; CD38$^+$/CD95$^-$) and GC B cells (CD38$^-$/CD95$^+$) are depicted in the plots. GC B cells (CD38$^-$/CD95$^+$) were further analyzed for GL-7 expression, compared to Fo.B cells and quantified.

C Quantification of Fo.B cells (top, percentages) and CD38$^-$/CD95$^+$/GL-7$^+$ GC cells (percentages and absolute numbers, bottom panels) in WT (black) and IgD$^{-/-}$ (green) animals after 4, 7, 10, and 14 days postimmunization, mean ± SD (numbers of animals indicated in the top plot). Statistical significance was analyzed by a two-tailed unpaired *t*-test.

D Analysis of GC in spleen sections from WT and IgD$^{-/-}$ animals after 4, 7, and 10 days postimmunization. Representative images of 2 × 2 mm sections stained with DAPI (nuclei blue) and peanut agglutinin (PNA red) are shown for the indicated time points. Scale bar represents 500 μm. Dashed white or yellow lines mark distinct GC foci in the merged images.

E Representative 800 × 800 μm regions of areas marked by yellow-dashed lines within the images of D are shown as enlarged single GCs. As no distinct GC foci were observed in IgD$^{-/-}$ samples from days 4 and 7, an identical 800 × 800 μm region is shown in each case.

F Quantification of number of GCs per mm$^2$ area of spleen sections from WT (black) and IgD$^{-/-}$ (green) animals after 4, 7, and 10 days postimmunization. Data represent mean ± SD of two sections (from three animals per group and time point) and were analyzed by two-tailed unpaired *t*-test.

G Total serum α-TNP IgG. Sera from PBS- or TNP-Ova-immunized WT (left) and IgD$^{-/-}$ (middle) mice, collected at day 7 following immunization, were adjusted to an IgG concentration of 1 μg/ml and applied in duplicates and dilution steps of 1:3 to TNP-coated plates, respectively, mean ± SEM. Right: Overlay of α-TNP IgG titers from TNP-Ova-immunized WT and IgD$^{-/-}$ mice. Statistical significance was calculated by using the Mann–Whitney *U*-test (see also Appendix Table S7), n.s. = not significant, *$P \leq 0.05$, **$P \leq 0.01$.

Furthermore, our results suggest that anergic B cells can only be detected *in vivo* when the pathways for receptor editing and clonal deletion of autoreactive B cells are blocked. In fact, blocking receptor editing by *Rag1* deficiency results in complete loss of *3-83$^{ki}$* autoreactive B cells (Halverson *et al*, 2004). Similar to this, loss of Pten abolishes the ability of B cells to undergo secondary gene rearrangements at the *IgL* gene locus. However, Pten deficiency rescues autoreactive B cells from clonal deletion and supports their survival as surface BCR-negative anergic B cells. It is conceivable that these anergic B cells lacking Pten have an appropriate strength of PI3K signaling required for B-cell survival (Srinivasan *et al*, 2009). Such anergic surface BCR-negative cells that are refractory to further external stimulation can also be detected in chronic lymphocytic leukemia (CLL; Stevenson & Caligaris-Cappio, 2004).

Notably, transgenic mice expressing only anti-HEL IgD were reported to induce Ca$^{2+}$ flux upon treatment with monomeric HEL (Sabouri *et al*, 2016). However, these results were only reported for mixtures of total splenic cells from anti-HEL IgD and anti-HEL IgM transgenic mice. It cannot be excluded that multimeric HEL complexes were present in these experiments. In addition, anti-HEL transgenic mice also secrete antibodies from endogenous *Ig* gene loci that might be involved in complex formation (Fig EV1A).

Our results are in complete agreement with previous data reporting a 3- to 4-day delay in the production of high-affinity antibodies in IgD-deficient mice suggesting that IgD "could confer a critical advantage in the defense against pathogens undergoing rapid expansion and mutational drift upon entry into the host" (Roes & Rajewsky, 1993). The finding that IgM- and IgD-deficient mice show comparable capacities to mount GC reactions reported by other studies (Nitschke *et al*, 1993; Noviski *et al*, 2018) may be attributed to differences in the immunization protocols such as the adjuvants used or the respective antigen (SRBC vs. TNP-Ova). TNP-Ova represents a multivalent antigen consisting of 10–20 TNP valences per Ova molecule. In contrast to this, SRBCs are entire cells endowed with an unlimited number of different antigens and epitopes (sugars, lipids, proteins, amino acids, and combinations thereof). Moreover, B-cell activation by SRBCs has been suggested to be mediated via PRRs and thus to be BCR-independent (Loetsch

*et al*, 2017). In fact, immunization with SRBCs does not require any additional adjuvants while immunization with TNP-Ova does. The finding that IgD-deficient mice show normal immune responses after immunization with SRBCs suggests that an immune response, which is sufficiently boosted by PRR ligands, is capable of activating the compromised IgD-deficient B cells, thereby masking the actual role of IgD for the immune system (DeFranco *et al*, 2012). Hence, the specific function of IgD in directing B cells into GCs and inducing efficient immune responses may become evident especially upon encounter with weak immunogens lacking typical pathogen-associated patterns involved in PRR activation. Stimulation via IgM, however, primarily leads to plasma cell differentiation and IgM secretion (Noviski *et al*, 2018; Setz *et al*, 2018).

Importantly, the delayed GC reaction in IgD-deficient mice demonstrated in the present study suggests that IgD is required for efficient initiation of GC formation, which is a prerequisite for affinity maturation (Shlomchik & Weisel, 2012; Bannard & Cyster, 2017).

Interestingly, experiments investigating antigen presentation revealed that, in contrast to polyvalent antigen, B cells show impaired presentation of monovalent antigen (Kim *et al*, 2006). Efficient antigen presentation is essential for B-/T-cell interaction and for subsequent T-cell help during the GC reaction where SHM results in affinity maturation (Jacob *et al*, 1991; Kim *et al*, 2006). In this context, it is conceivable that expression of IgD BCR is an important element for the execution of a rapid and efficient humoral immune response that provides an evolutionary advantage for the control of an acute infection. Thus, it should be no surprise that IgD expression is a conserved feature of most vertebrate species.

Although IgD is later downregulated in GC B cells (Jacob *et al*, 1991), the data suggest that IgD is optimized for recognition of antigen complexes and for efficient antigen presentation to T cells, thereby facilitating the recruitment of B cells into GCs and the efficient maturation of antibody responses. The fact that FoxO1, which has recently been shown to be essential for the GC reaction, activates IgD expression is in full agreement with this scenario

(Dominguez-Sola *et al*, 2015; Sander *et al*, 2015). Our findings further suggest that FoxO1 controls the expression of *Ell2* (Martincic *et al*, 2009), which regulates the recognition of the poly-adenylation sites upstream of the δHC exons, and of the splicing factor *Zfp318* that activates the splicing of the rearranged *VDJ* cassette to the δHC exons in the pre-mRNA (Enders *et al*, 2014).

Importantly, our data suggest that IgD is not part of an "anergy response" that results in functional silencing of autoreactive B cells. In contrast to the herein reported anergic autoreactive B cells that lack surface BCR expression, IgD-expressing B cells respond to antigen complexes and to anti-BCR treatment (Cooke *et al*, 1994; Ubelhart *et al*, 2015). Thus, referring to IgD-expressing B cells as "anergic" confuses B-cell anergy with maturation. IgD expression marks B-cell maturation and collaborative regulation of immune responses. In fact, by responding to immune complexes formed by other components of the immune system, IgD-expressing B cells are optimized for the presentation of antigen to T helper cells resulting in efficient recruitment of B cells into GCs and subsequent maturation of antibody responses. In this scenario, IgD BCR seems to act as a link between innate components of the immune system, such as complement factors and natural IgM, and adaptive components represented by high-affinity antibodies.

Thus, elucidating IgD function may not only advance our understanding of mature B-cell development and function but also lead to improved vaccination protocols by controlled delivery of antigen to IgD-expressing B cells during immunization.

# Materials and Methods

### Mice

*Pten^{f/f}* mice (provided by T. Mak (Suzuki *et al*, 2001), Campbell Family Institute for Breast Cancer Research at Princess Margaret Hospital, Toronto, ON, Canada), were crossed to mb1-cre (CD79a-cre) transgenic mice (Hobeika *et al*, 2006) to achieve B-cell-specific cre-mediated recombination in early pro-B cells. *Pten^{f/f}* × mb1-cre mice were bred to *3-83* knock-in mice (either on H2-K$^b$ background or on H2-K$^d$ background; Pelanda *et al*, 1997) or *MD4*-transgenic mice (*MD4^{tg}*); *MD4* × *ML5* (*ML5^{tg}*: hen-egg lysozyme (HEL); Goodnow *et al*, 1988). *FoxO1^{f/f}* mice (Tothova *et al*, 2007) were described earlier. Mice analyzed in this study were sacrificed at the age of 6–8 weeks.

10- to 15-week-old WT and IgD$^{-/-}$ mice (Nitschke *et al*, 1993) were immunized intraperitoneally (i.p.) with a mixture of 100 µg of 2,4,6-trinitrophenyl hapten conjugated to ovalbumin (TNP-Ova comprising 10–20 TNP epitopes per Ova molecule; Biosearch) and Alu-S-gel (Alum; Serva) or 10% sheep red blood cells (SRBC; Cedarlane Labs), while control mice obtained PBS + alum. Animal experiments were performed in compliance with license 1,288 for animal testing at the responsible regional board Tübingen, Germany.

All mice used in this study were bred and housed in the animal facility of Ulm University under specific-pathogen-free conditions. All animal experiments were done in compliance with the guidelines of the German law and were approved by the Animal Care and Use Committees of Ulm University and the local government.

### Cell culture and biochemistry

Ramos cells were cultivated in RPMI (Gibco) while Phoenix and mature splenic B cells were cultured in Iscove's medium (Biochrom AG). The media were supplemented with 5% heat-inactivated FCS (Sigma), 2 mM L-glutamine (Gibco), 100 U/ml penicillin/streptomycin (Gibco), and 50 µM β-mercaptoethanol, respectively. For survival assessment of anergic B cells, 2.5 µg/ml lipopolysaccharides (LPS, Sigma) and/or 10 ng/ml murine IL-4 (immunotools) were added to the medium.

### Plasmids and retroviral transduction

The retroviral expression vectors encoding the constitutively active FoxO1 (FoxO1-A3, harboring the point mutations T24A, S254A, and S319A at the Akt phosphorylation sites; Alkhatib *et al*, 2012) or cre (Abdelrasoul *et al*, 2018), respectively, were generated as previously described. Immunoglobulin HCs and LCs were expressed by using the BiFC vector system as described (Kohler *et al*, 2008). Viral supernatants were generated using the Phoenix retroviral producer cell line as described in the manufacturer's instructions. Phoenix cells were cultured for 48 h in Iscove's culture medium + 5% FCS. Cells were plated at a density of $2 \times 10^5$ cells/ml to generate supernatants by using the transfection reagent GeneJuice (Merck Millipore). Retroviral supernatants were harvested after 48 h. Mature splenic mature B cells were pre-treated with 2.5 µg/ml LPS (Sigma). For transduction, the cells were mixed with supernatants and centrifuged at 300 *g* and 37°C for 3 h, whereas Ramos cells were cultured overnight in retroviral supernatants. Transduced cells were returned to culture for at least 3 days before analysis.

### Calcium measurement

Multivalent cHEL was generated as previously described by biotinylation of HEL (Ubelhart *et al*, 2015) and crosslinking with unconjugated streptavidin (Invitrogen) at a ratio of 1:2. Intracellular $Ca^{2+}$ mobilization was measured as described (Storch *et al*, 2007; Ubelhart *et al*, 2015). For measurement of $Ca^{2+}$ mobilization cells were loaded with the $Ca^{2+}$-sensitive dye Indo-1 (Molecular Probes; Invitrogen) and stimulated with HEL (Sigma) at the indicated concentrations (the amount of HEL was constant in both HEL antigen configurations), BSA (Serva) and 10 µg/ml α-mouse κLC (polyclonal; Southern Biotech). WT Ramos cells expressing endogenous BCR were stimulated with 10 µg/ml α-human λLC (polyclonal; Southern Biotech) or 10 µg/ml α-human µHC antibody (polyclonal; Southern Biotech), respectively. For analysis of $Ca^{2+}$ mobilization in splenocytes, total splenic B cells were pre-enriched by using the B-cell isolation kit, mouse or the Pan-B-cell isolation kit II, mouse (both from Miltenyi Biotec) for Pten-deficient cells according to the manufacturer's instructions. To exclude residual non-B cells, purified cells were stained with α-CD90.2-PE (53-2.1; BD) prior to loading with Indo-1. $Ca^{2+}$ flux measurements were acquired at a FACS LSR Fortessa flow cytometer (BD).

### Flow cytometry

Cell suspensions were blocked with α-CD16/CD32 Fc-Block (2,4G2; BD) and stained by standard procedures. Intracellular flow cytometric staining was performed by using FIX & PERM Cell Fixation and

Permeabilization Kit (ADG Nordic-MUbio). Cells were stained using the antibodies enlisted in Appendix Table S8.

Biotin was detected by using streptavidin Qdot605 (Molecular Probes; Invitrogen) or α-biotin-PE (1D4-C5, BioLegend). The 54.1 antibody recognizing the 3-83 idiotype was kindly provided by Roberta Pelanda and David Nemazee. Viable cells were distinguished from dead cells by staining with Fixable Viability Dye eFluor 780 (eBioscience). Cells were acquired at a FACS Canto II flow cytometer (BD). If not stated otherwise numbers in the dot plots indicate percentages in the respective gates while numbers in histogram plots state the mean fluorescence intensity (MFI).

### Immunohistochemistry

Spleens were embedded in OCT compound (SAKURA) and frozen at −80°C. 5 μm sections were prepared using a cryo-microtome (Reichert-Jung 2800 Frigocut) with a S35 knife (Feather) and fixed on SuperFrost Plus slides (Thermo Scientific) by treatment with pure acetone.

For generating overviews of spleens, sections were rehydrated with PBS + 2% BSA + 0.1% Na-azide and blocked with Fc-Block (α-CD16/32; BD Biosciences).

For detection of germinal centers, spleens were fixed in 4% paraformaldehyde (PFA, Santa Cruz) for 3–4 h at 4°C and subsequently dehydrated in 30% sucrose. Sections were fixed with (95:5) acetone/methanol (Merck) and afterwards blocked with 2.5% rabbit and 2.5% mouse serum (vector laboratories). Sections were stained with antibodies enlisted in Appendix Table S9 and mounted with fluoromount-G containing 4′,6′-diamidino-2-phenylindole (DAPI; Southern Biotech). Stained sections were analyzed using fluorescence microscopes Axioskop 2 (Zeiss) or DMi8 (Leica).

### Analysis of *LC* gene recombination

Genomic DNA was isolated from purified splenic B cells of $Pten^{f/f} \times 3\text{-}83^{ki}$ and $Pten^{f/f} \times mb1\text{-}cre \times 3\text{-}83^{ki}$ on the indicated backgrounds (H-2K$^b$ or H-2K$^d$). PCRs were performed using the primers enlisted in Appendix Table S10.

### Quantitative reverse transcriptase (qRT-)PCR

Cells were purified using a FACS Aria IIu sorter (BD). RNA was extracted from FACS-sorted cells by using the ReliaPrep RNA Cell Miniprep System (Promega). Residual genomic DNA was digested using DNase I (Thermo Fisher). cDNA synthesis was performed with the RevertAid Reverse Transcriptase (RT) kit (Thermo Fisher) as indicated by the manufacturer. Removal of genomic DNA was verified by PCR on RNA samples subjected to cDNA synthesis in absence of RT. Quantitative (q)PCR analysis was performed by using the TaqMan-probe mixes (Applied Biosystems), enlisted in Appendix Table S11, together with TaqMan-gene expression mastermix (Applied Biosystems). qPCR data were acquired on a StepOnePlus real-time thermocycler (Applied Biosystems) in triplicates and analyzed with the StepOne Software v2.3. Results were calculated by applying the $\Delta\Delta C_T$-method.

### Enzyme-linked immunosorbent assay

96-well plates (NUNC, Maxisorp) were coated either with polyclonal α-mouse IgM or IgG antibody (SouthernBiotech), respectively, and blocked with buffer containing 1% BSA. Dilutions of mouse IgM or IgG antibodies (SouthernBiotech) were used as standard. The concentration of IgM or IgG antibodies in the sera was determined by detection with alkaline phosphatase-labeled α-mouse IgM or IgG (Southern Biotech), respectively. P-nitrophenylphosphate (Genaxxon) in diethanolamine buffer was added, and data were acquired at 405 nm using a Multiskan FC ELISA plate reader (Thermo Scientific). For detection of α-HEL antibodies, sera were adjusted to an IgM concentration of 500 μg/ml and applied in dilution steps of 1:3 to plates coated with 10 μg/ml HEL. For detection of α-TNP IgG antibodies, sera were adjusted to an IgG concentration of 1 μg/ml and applied in dilution steps of 1:3 to plates coated with 1 μg/ml TNP-conjugated BSA.

### Statistical analysis

Graphs were created, and statistical analysis was performed by using GraphPad Prism (version 6.0h) software. The numbers of individual replicates or mice ($n$) are stated in the figure or figure legends. $P$ values were calculated by the tests stated in the respective figure legends. $P$ values < 0.05 were considered to be statistically significant (n.s. = not significant; $*P \leq 0.05$; $**P \leq 0.01$; $***P \leq 0.001$; $****P \leq 0.0001$).

**Expanded View** for this article is available online.

### Acknowledgements

We thank A. Tietz and N. Gust for technical help in flow cytometric analyses and ELISA, G. Allies for cell sorting, T. Wossning for the analysis of receptor editing in *3-83$^{tg}$* mice by PCR, R. Übelhart for assistance in BCR expression experiments using Ramos cells, P. Möller for providing access to the microtome, T. Mertens, T. Stamminger, and J. von Einem for access to the fluorescence microscope, and A. Ushmorov for access to the sonicator. C. Murre and Y. Lin kindly provided FoxO1 ChIP-Seq data. This work was supported by the DFG through TRR130 (B cells and beyond) projects P01, P02, P04, and P08, SFB1074 (Experimental Models and Clinical Translation in Leukemia), SFB1279 (Exploration of the Human Peptidome), EXC294, and ERC advanced grants 694992 and 322972 for HJ and MR, respectively.

### Author contributions

CSS analyzed the *3-83$^{ki}$* and *MD4$^{tg}$* mice, performed the transduction experiments of splenic B cells, analyzed data, prepared the figures together with PCM, and contributed to writing of the manuscript. LN generated the IgD$^{−/−}$ mice. PCM, AK, VR, and EG analyzed GC formation in IgD$^{−/−}$ mice. JW, MR, and XH established the HL-KO Ramos cells that were transfected and analyzed by JI and VR together with AK. MY analyzed the ChIP-Seq data. MD assisted with the ChIP assay. HJ designed the study, proposed the experiments, supervised the work, and wrote the manuscript. All co-authors read and discussed the manuscript.

### Conflict of interest

The authors declare that they have no conflict of interest.

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
