## [Review Process File · The EMBO Journal]

Pten controls B cell responsiveness and germinal center reaction by regulating the expression of IgD BCR

Corinna S. Setz, Ahmad Khadour, Valerio Renna, Joseena Iype, Eva Gentner, Xiaocui He, Moumita Datta, Marc Young, Lars Nitschke, Jürgen Wienands, Palash C. Maity, Michael Reth, Hassan Jumaa

Review timeline:	Submission date:	12th Jul 2018
	Editorial Decision:	8th Aug 2018
	Revision received:	8th Dec 2018
	Editorial Decision:	31st Jan 2019
	Revision received:	13th Mar 2019
	Editorial Decision:	21st Mar 2019
	Revision received:	22nd Mar 2019
	Accepted:	25th Mar 2019

Editor: Karin Dumstrei

Transaction Report:

1st Editorial Decision

8th Aug 2018

Thank you for submitting your manuscript to The EMBO Journal. Your study has now been seen by three referees and their comments are provided below.

As you can see from the comments below the referees find the analysis interesting. However, they also raise a number of constructive and good points that should be addressed in order to consider publication here. All three referees find that the immunisation experiments in Figure 7 needs to be improved. In addition more data is needed to support if Pten deficiency impacts B cell differentiation or simply impairs the expression of markers IgD and CD23.

Should you be able to address the raised concerns in full then I would like to invite you to submit a revised manuscript. I should add that it is EMBO Journal policy to allow only one single round of major revision, and that is is therefore important to address the major points raised at this stage.

REFeree REPORTS:

Referee #1:

The regulation and function of IgD receptors (IgD-BCRs) in mature B cells remain poorly understood, mostly because these cells co-express surface IgM receptors that are largely sufficient to implement B cell development, survival and activation. In this manuscript, Setz et al. show that the lipid phosphatase PTEN up-regulated IgD-BCR expression by inhibiting PI3-kinase, a signal-transducer that represses the activation of the transcription factor FoxO1. Consequently, PTEN-deficient B cells showed reduced IgD expression, whereas B cells overexpressing a constitutively active form of FoxO1 exhibited increased IgD-BCR expression. In agreement with earlier elegant

studies published by the same group, IgD-BCR rendered B cells more responsive to multivalent antigens. Accordingly, IgD-deficient B cells mounted delayed germinal center-mediated T cell-dependent responses to immunocomplexes upon immunization. It is concluded that the PI3-kinase-FoxO1 axis is critical for the activation of IgD expression, which in turn is instrumental to generate mature B cells fully responsive to antigen complexes.

General comment

This is an elegantly presented, rigorous and well-written manuscript that significantly extends our knowledge on the regulation and function of IgD-BCR in mature B cells. The authors used a clever mix of state-of-the-art *in vivo* and *in vitro* experimental systems to analyze the contribution of PTEN, PI3-K and FoxO1 to IgD-BCR expression and to dissect the impact of multivalent vs monovalent antigen on IgD-BCR-mediated B cell activation. The following specific comments are provided to further enhance the manuscript.

Specific comments

- 1) Does IgD-BCR show the same properties in splenic follicular and marginal zone B cells?
- 2) Besides triggering calcium fluxes, does multivalent antigen preferentially elicit B cell proliferation or antigen presentation through IgD-BCR?
- 3) Does IgD-BCR engagement by multivalent or soluble T cell-dependent antigens results in a different quality of the Bcl6-dependent germinal center program? For example, is affinity maturation simply delayed or also qualitatively perturbed in IgD-deficient mice?
- 4) IgD-BCR deficiency clearly alters the dynamics of the germinal center response, causing its delay. Is this delay due to defective proliferation, survival or (Bcl6-dependent) differentiation of IgD-deficient germinal center B cells? And does this delay have any impact on a humoral response against an invading microbe?

Referee #2:

Naïve B lymphocytes express their antigen receptor (BCR) in two forms, as membrane IgM and membrane IgD. The experiments presented in this manuscript address several loosely related questions related to the unique function of IgD in mouse and human, and how is its expression regulated. While a number of important issues are addressed in this manuscript, in the opinion of this reviewer, each of these issues is addressed in a limited way that is insufficient to provide substantial, novel insights that would contribute importantly to the field. The following comments give my thoughts about the limitations of what can be concluded regarding four specific conceptual issues addressed by the experiments.

1) Role of FoxO1 in controlling the expression of membrane IgD: One potentially significant conclusion relates to the role of the transcription factor FoxO1 in controlling the expression on the cell surface of the IgD isoform of the BCR. The initial experiments showed that deletion of PTEN in B cells in an IgH + IgL transgenic setting (3-83 transgenic mouse) leads to a defect in expression of IgD on the B cells, and a defect of receptor editing via Ig kappa rearrangement, a process which is prominent with this transgene even in the absence of the nominal antigen (H2 b allele of the H2 locus). As PTEN acts by removing a phosphate from the signaling lipid mediator PIP3, and PIP3 acts to inhibit the transcription factor FoxO1 (via phosphorylation by Akt), the net effect of deletion of PTEN is to increase PIP3 signaling, leading to inactivation of FoxO1. Therefore, the authors used gain and loss of function experiments to assess whether FoxO1 was responsible for IgD expression in the normal setting. In the text of the manuscript, the authors indicate (correctly) that the effect on IgD expression could reflect regulation by FoxO1 of genes known to regulate IgD expression (especially *Zfp318*) OR that it could reflect a requirement for FoxO1 for B cells to transition from an immature state, which fails to express IgD to a mature state, which does express IgD. This possible interpretation is suggested by the fact that FoxO1 is shown to also regulate CD23, another cell surface protein that is expressed in mature B cells but not immature B cells. Both of these possible mechanisms are discussed in the text of the ms., the applicants fail to distinguish between

them in a compelling way, as for example might be achieved by RNA-Seq and/or by CHIP of FoxO1 to possibly detect binding to the promoters of Zfp318 or CD23. Given the methodology used for introduction of the constitutively active mutant FoxO1 or the Cre (to delete floxed allele of FoxO1) in the presented experiments, the regulation of expression seems more likely than the developmental block model, but more direct experiments are needed to address this point in a sufficiently compelling manner. If FoxO1 is regulating IgD expression independently of maturation state, that might explain why IgD is rapidly downregulated upon activation of mature B cells and is expressed at a low level on germinal center B cells (as FoxO1 would presumably be turned off by elevated PIP3/Akt in these circumstances). This would be a significant insight for the field.

2) Role of PTEN and IgD in B cell anergy (a cellular mechanism that promotes immune tolerance to self antigens): The authors present evidence that MD4 Ig transgenic B cells and 3-83 Ig transgenic B cells still achieve inactivation to self antigens by "anergy" in the absence of PTEN. These results disagree with a major conclusion published previously with the MD4 Ig transgenic B cells by Brown et al. (2009) (Ref in the manuscript). This is an important issue, but in my opinion incompletely addressed in the ms. Having said that, the authors do have some interesting observations and further analysis could be highly informative. The data shown make the case that in the 3-83 Ig transgenic B cells, anergy can still be achieved in the absence of PTEN by severe downregulation of both forms of the BCR: IgD because it requires FoxO1 activity and IgM perhaps by antigen-induced internalization. Characterization of anergy is minimal, the authors examine the lack of BCR-stimulation-induced calcium elevation, but more cellular aspects of anergy are not examined. Do these cells have a short half-life *in vivo*? If T cell help is supplied, do they have a poor antibody response compared to non-nergic 3-83 B cells deleted for PTEN? For the experiments with MD4 Ig transgenic mice (which have an especially well-characterized anergy), again characterization of the anergic phenotype is minimal (less decreased BCR-induced calcium increase and unaffected secretion of antigen-specific IgM (without overt immunization). Previous work by Browne et al involved a considerably more extensive characterization of anergy and found that PTEN was required for a number of anergy-related properties, such as rapid death upon acute exposure to antigen. In short, with a more extensive characterization of anergy or lack thereof in these two Ig transgenic systems following PTEN-deletion, there likely could be a fine contribution. (Note: the different Ig transgenic systems in which anergy occurs have a range of phenotypes, possibly reflecting shallower to deeper forms of anergy, so it is worthwhile to measure parameters related to anergy in both systems and to what extent they still occur following deletion of PTEN).

3) Differential response of mIgD vs. mIgM to monomeric vs. oligomeric forms of anergy: This conceptual issue follows on the authors' fine previous publications and extends those observations in significant ways, mainly showing that human IgD behaves very similarly to murine IgD, and that murine IgD shows this differential effect in a transfected human B cell line. While of value to the field, in a conceptual sense, these studies extend the previous work in an incremental way rather than blazing a new trail conceptually.

4) Role of IgD in the germinal center response (Figure 7): The overall conclusion reached in this ms. that the germinal center response of IgD^{-/-} B cells is considerably delayed disagrees with results of a recent publication (Noviski et al 2018, Ref in manuscript). This could be due to experimental differences of various types. In the opinion of this reviewer, a more extensive analysis would be needed here to really provide useful insights, including a demonstration of a cell-intrinsic effect (e.g., using adoptive transfer or mixed bone marrow chimeric mice), since there could be cell extrinsic effects due to secreted antibody binding to the antigen and IgD deletion may alter the antibody secreted from the non-germinal center component of the antibody response.

Minor comments:

1) The title of the manuscript overstates what can be concluded from the data presented (PTEN deletion likely has effects on B cells responsiveness that go beyond just regulation of IgD expression).

2) Figure 4: the flow cytometry here should include appropriate negative control antibody staining profiles.

3) Title to Figure 6 is an overstatement of what can be reasonably concluded here, these may be two distinct effects of PTEN deficiency, rather than one being dependent on the other. Note also that PTEN deficiency increases the calcium response to BCR crosslinking even in a non-nergic setting, likely reflecting Btk and/or PLC-gamma2 binding to PIP3 in the plasma membrane as one parameter controlling their activation.

Referee #3:

This manuscript is part of a series of work from the Jumaa lab investigating both the PI3K/Pten pathway and the immunological functions of IgD. They show in excellent genetic experiments that FoxO1 is required for IgD+ B cells and that blocking FoxO1 upregulation by overactive PI3K (through Pten deficiency) results in reduced IgD, among other consequences. Using a genetic complementation system in a human B cell lines, the authors show that murine and partially humanised anti-HEL IgM responds well to both monovalent and multivalent Ag, while equivalent IgD receptors only respond to multivalent Ag. This is a confirmation of earlier work by the lab in another experimental model. Compatible results were obtained in mouse cells carrying transgenic BCRs +/- Pten. Finally the importance of IgD was tested in a standard immunization model and revealed a defect in early GC responses.

Overall this is a solid study that addresses an interesting question. The data are very clearly presented and the experimentation and analysis generally appropriate. Some aspects including figure 5-6 confirm previous data from the lab, albeit in a new model. Moreover the data in figure 7, which is used to conclude that IgD is important early in the GC response, does contradict a recent paper by another group (Noviski eLife 2018). More work is needed to clarify this issue, as it is key to the authors' overarching model.

Specific major concerns

1. The authors show reduced IgD and CD23 in the absence of Pten and conclude that Pten regulates the generation of follicular B cells line 200-202. The alternative is that these markers are simply deregulated, which a conclusion that may be supported by the regulation of splicing factors for IgD by FoxO1 and the relatively normal numbers of splenic B cells (figure 2E, S2). Some histology and staining of spleen and Lymph nodes would test if the normal follicular structures are present.
2. Figure 4 and S4. Figure S4 shows that caFOXO1 expression results in more IgD mRNA versus controls. As IgD is a splice variant this analysis isn't really relevant as this could represent a change in IgH transcription in total. The key issue is to determine the proportion of total IgH transcripts that are IgD or IgM. In this context, it is also notable that FOXO loss (Figure 4E) appears to result in less surface IgM staining; this should also be quantified as a control. Finally, I find the CD23 rescue unconvincing. The change in Figure 4B is very marginal and much below the WT situation. At the least CD23 mRNA expression here (and in figure 4E) should be measured using appropriate positive and negative controls as this phenomenon is predicted to be transcriptional.
3. Figure 6. While the trends seem to fit with the authors' general model, the relatively small changes in Ca flux need to be quantified over multiple experiments to determine how robust they are.
4. Figure 7. I have several concerns about this data, particularly as it is important to the authors overall conclusion and disagrees with the results of SRBC immunization of IgD^{-/-} mice shown in Noviski (eLife 2018). The GCB cell quantitation in panel also appears problematic, as the GC reaction seems both modest and transient, with a low peak in the WT at day 7. This appears a suboptimal response using NP-OVA in Alum. This experiment would be much more convincing if the authors repeated the analysis and tracked the frequency of Ag specific NP-binding GCB cells, instead of the surrogate markers used.

Point by point response

Referee #1:

Reviewer 1 considered our study to be „elegantly presented, rigorous and well-written“ and points out that our findings „significantly extend our knowledge on regulation and function of IgD-BCR in mature B cells“. We thank the reviewer for her/his comments, which were addressed as described below. Changes in the manuscript text, referring to comments by reviewer #1 are marked in yellow.

1) *Does IgD-BCR show the same properties in splenic follicular and marginal zone B cells?*

To address this question, we stimulated splenic B cells from MD4^{tg} mice with sHEL, cHEL or α - κ LC antibodies, respectively and gated on CD21^{hi}/CD23^{lo/-} (marginal zone (MZ.B)) and CD21^{lo}/CD23⁺ (follicular (Fo.B)) B cells. The data are shown in Fig. S6C-E.

Staining for surface expression of IgM, IgD and Ig- κ LC revealed that Fo.B cells express higher levels of IgD and slightly lower amounts of IgM compared to MZ.B cells (Fig. S6C) suggesting that Fo.B cells may respond better to complex than to soluble antigen. Surface expression of κ LC and HEL-binding capacity was largely comparable in both subpopulations. Our data show that upon exposure to either stimulus, the Ca²⁺ response of Fo.B cells appears to be weaker as compared with MZ.B cells (Fig. S6D-E). Notably, besides the differential IgM/IgD expression, MZ.B and Fo.B also differ in expression of other markers including CD21 and CD23. Moreover, Fo.B cells still express considerable amounts of IgM, which is not the case in Ramos cells reconstituted with HH10-IgD, or in MD4^{tg} mice that drastically down-regulate IgM-expression in the presence of the cognate antigen (page 15, lines 312 – 316).

2) *Besides triggering calcium fluxes, does multivalent antigen preferentially elicit B cell proliferation or antigen presentation through IgD-BCR?*

Based on our findings, it is tempting to speculate that antigen binding and signaling via IgM- and IgD-BCR leads to further differences in down-stream functions such as antigen presentation or proliferation.

Based on previous data showing that stimulation of B cells with multivalent antigen results in efficient antigen presentation as compared to stimulation with monovalent antigen (Kim et al., 2006), we hypothesize that complex antigens bound and internalized by IgD-BCR might be directed into the exogenous pathway of antigen-presentation, resulting in peptide presentation complexed with MHC class II molecules. Readouts addressing the interaction of IgD-BCR with MHC class II molecules and subsequent presentation to cognate T cells require complex experimental systems that are currently being established at our lab.

3) Does IgD-BCR engagement by multivalent or soluble T cell-dependent antigens result in a different quality of the Bcl6-dependent germinal center program? For example, is affinity maturation simply delayed or also qualitatively perturbed in IgD-deficient mice?

To investigate the mechanism by which IgD promotes recruitment of B cells into GC reaction, we analyzed Bcl6 expression in GC B cells from immunized WT and IgD^{-/-} mice. In these experiments (see figure below), we could not detect any difference in Bcl6 induction between WT and IgD^{-/-} GC B cells, neither upon immunization with TNP-Ova, nor with SRBC. However, we cannot exclude that differences in Bcl6 induction occur at earlier time points (before day 7) of the immunization. Therefore, we did not include these data in the revised version of the manuscript.

Figure for Reviewer #1, comment 3: Bcl6 in WT and IgD^{-/-} GC B cells (related to Fig. 7)

A | WT and IgD^{-/-} mice were immunized either with TNP-Ova, SRBC or treated with PBS as control and analyzed after 7 days for GC formation as shown in Fig. 7B. GC and non-GC B cells were stained intracellularly for Bcl6 expression. Numbers in the histograms indicate percentages of Bcl6⁺ cells.

B | Quantification of Bcl6⁺ GC B cells in WT and IgD^{-/-} mice (from A) immunized either with TNP-Ova, SRBC or treated with PBS. Single dots represent data from individual mice.

4) *IgD-BCR deficiency clearly alters the dynamics of the germinal center response, causing its delay. Is this delay due to defective proliferation, survival or (Bcl6-dependent) differentiation of IgD-deficient germinal center B cells? And does this delay have any impact on a humoral response against an invading microbe?*

This question is also linked to comment 3 raised by the same reviewer.

To characterize the underlying molecular mechanisms, which may be involved in the IgD-dependent GC-response, we analyzed Bcl6 expression and proliferation (by staining for Ki-67, see Figure below) in TNP-Ova-immunized GC B cells from WT and IgD-deficient mice. Our data suggest that neither Bcl6 expression (see comment 3 by reviewer #1) nor proliferation differ between GC B cells from either genotype. As already pointed out for the Bcl6 data, it is not excluded that differences in Ki-67 might occur at earlier time points of the immunization. Therefore, we did not include the data in the revised manuscript.

To assess if the observed delay in the GC-response may affect the humoral immune response, we measured the α -TNP total IgG content in sera from immunized WT and IgD^{-/-} mice. We detected lower titers of antigen-specific IgG in sera from IgD^{-/-} mice as compared to WT suggesting that the delayed GC-response also results in reduced production of class-switched and affinity-matured antibody (results are shown in Fig. 7G; page 16, lines 344 – 346; page 45, lines 1000 - 1005).

Figure for Reviewer #1, comment 4: Ki-67 in WT and IgD^{-/-} GC B cells (related to Fig. 7)

A | WT and IgD^{-/-} mice were immunized with TNP-Ova and analyzed after 7 days for GC formation as shown in Fig. 7B. GC and non-GC B cells were stained intracellularly for Ki-67. Numbers in the histograms indicate the MFI.

B | Quantification of Ki-67 mean fluorescence intensity (MFI) in GC and non-GC B cells in WT and IgD^{-/-} mice (from A) immunized either with TNP-Ova.

Referee #2:

According to reviewer #2 our manuscript addresses questions in a “limited way limited way that is insufficient to provide substantial, novel insights that would contribute importantly to the field.” Nevertheless reviewer #2 states that the study reports “some interesting observations and further analysis could be highly informative”. We thank the reviewer for her/his comments, which were addressed as described below. Changes in the manuscript text, referring to comments by reviewer #2 are marked in green.

1) Reviewer #2 requested further data investigating whether FoxO1 has a direct effect on the regulation of IgD or if Pten-deficiency merely retains the cells at a developmental stage at which they cannot upregulate maturation markers such as IgD or CD23. Therefore, reviewer #2 asked for data addressing the question whether there is direct binding of FoxO1 to the genes regulating IgD expression (Zfp318 and Ell2) and CD23. According to reviewer #2 this could explain why IgD is rapidly downregulated upon activation of mature B cells and why it is expressed at a low level on germinal center B cells.

This comment is also connected to point 1 raised by reviewer #3.

To distinguish whether Pten-deficiency impairs developmental progression to a stage at which B cells can upregulate maturation markers IgD and CD23 or if lack of Pten directly affects regulation of these markers, we performed immunohistochemistry on spleen sections from *Pten^{fl/fl}*, *Pten^{fl/fl} x mb1-cre* mice and *Pten*-deficient mice with pre-rearranged Ig gene cassettes (Fig. 3E). Our data show abnormal follicles in the spleens of conditional *Pten*-deficient mice. In contrast to spleens from WT mice, spleens from *Pten^{fl/fl} x mb1-cre* mice show no organized structures for B cells (Fig. 3E & S2) (Setz et al., 2018). Introducing pre-rearranged Ig genes increases the numbers of splenic B cells. However, the population of follicular B cells residing in the area between the marginal zone and T cell zone is smaller in the absence of Pten as compared with WT controls. Therefore, we conclude that the development of mature Fo.B cells requires Pten expression (page 10, lines 198 – 209; page 40, lines 890 - 895).

To investigate whether FoxO1 regulates IgD (and CD23) expression by directly binding to gene loci encoding *Zfp318*, *Ell2* and *Fcer2*, we first analyzed available data on genome-wide FoxO1 occupancy in B cells (Lin et al., 2010). These ChIP-Seq data, chromatin immunoprecipitation combined with deep DNA sequencing, showed no FoxO1 binding-sites within the genes of interest (page 12, lines 254 – 256). In addition, we designed a chromatin ChIP assay (data are shown in Fig. S5). To this end, we screened the target gene sequences for the presence of the consensus motif for FoxO1 binding (Fig. S5A) (Barthel et al., 2005; Furuyama et al., 2000) and for sequences with similarities to two FoxO1 binding motifs identified in the Pax5 gene (Lin et al., 2010) (Fig. S5B). We detected 11 putative FoxO1 binding sites in the *Zfp318* locus, 17 in *Ell2* and 7 in *Fcer2* (encoding CD23) (Fig. S5C). For precipitation we used an antibody that was already used for FoxO1 ChIP (Shin et al., 2012). Similar to the available data (Lin et al., 2010), we could not detect any evidence for direct binding of FoxO1 in these genes (Fig. S5D; pages 12 – 13, lines 257 – 262). Taken together, our data suggest that FoxO1 indirectly regulates *Zfp318*, *Ell2* and *Fcer2* (encoding CD23) (page 12, lines 254 – 256; pages 12 – 13, lines 257 – 262, 264).

2) According to reviewer #2 the discrepancy between our findings, showing that anergy is possible in absence of Pten and published data is “incompletely addressed”. Reviewer #2 states that anergy is poorly characterized: He/she would like to know whether these cells have a shortened half-life in vivo and they exhibit a poor antibody response in comparison to non-anergic cells. Reviewer #2 claims that previous work by Browne et al. involved a considerably more extensive characterization of anergy and found that Pten was required for a number of anergy-related properties, such as rapid death upon acute exposure to antigen.

Our data suggest that anergic B cells can only be detected *in vivo* when the pathways for receptor editing and clonal deletion of autoreactive B cells are blocked. In fact, blocking receptor editing by Rag1-deficiency results in complete loss of 3-83^{ki} autoreactive B cells (Halverson et al., 2004). Similar to Rag-1 deficiency, loss of Pten in our study abolishes the ability of B cells to undergo secondary gene rearrangements at the LC gene locus. However, Pten deficiency rescues autoreactive B cells from clonal deletion and supports their survival as surface

BCR-negative anergic B cells. It is conceivable that these anergic B cells lacking Pten have an appropriate strength of PI3K signaling required for B cell survival (Srinivasan et al., 2009).

Since we can only detect anergic cells in a system in which the pathway for their immediate elimination is interrupted, we would expect the half-life of such anergic cells to be comparable to that of wild-type mature B cells.

To further test this, we compared survival of Pten-deficient 3-83^{ki} B cells on H2-K^b (+Ag) and H2-K^d background upon stimulation with LPS, IL-4 or a combination of both factors (Fig. S1B). We detected no differences in the survival of untreated anergic and non-anergic B cells. Stimulation with IL-4 initially increased survival of both B cells in presence and absence of antigen. This effect was particularly pronounced in anergic B cells after 2 days of stimulation. No significant difference regarding survival between both groups was detected at later time points. LPS treatment (also in combination with IL-4) significantly increased survival of both anergic and non-anergic cells. Interestingly, in all treatment conditions anergic B cells exhibit a trend towards a prolonged survival as compared with B cells on the non-autoreactive background, suggesting that these anergic B cells lacking Pten do not have a shortened life span (page 9, lines 171 – 174; pages 20 – 21, lines 413 - 422). This is further supported by the finding that there is no significant difference in the absolute numbers of B cells in spleens from Pten-deficient 3-83^{ki} B cells mice in presence and absence of antigen (Fig. 1D, Tables S2-3).

Serum IgM concentrations are significantly reduced in *Pten*^{ff} x mb1-cre x 3-83^{ki} mice on the autoreactive background indicating that antibody responses are impaired in B cells from the respective mice (Fig. 1C; pages 8 – 9, lines 167 – 171).

3) Differential response of mIgD vs. mIgM to monomeric vs. oligomeric forms of energy: This conceptual issue follows on the authors' fine previous publications and extends those observations in significant ways, mainly showing that human IgD behaves very similarly to murine IgD, and that murine IgD shows this differential effect in a transfected human B cell line. While of value to the field, in a conceptual sense, these studies extend the previous work in an incremental way rather than blazing a new trail conceptually.

To show that the selective responsiveness of IgD as compared to IgM is not an

artifact of the experimental system using TKO pro-B cells and inducible SLP65/BLNK (Ubelhart et al., 2015), we used Ramos cells to establish an independent system with mature B cell characteristics and endogenous signaling machinery.

As the function of IgD BCR on mature B cells still remains elusive and subject to controversial discussion, we believe that the new data using the Ramos cells are important. In particular, these data show without doubt that the difference in responsiveness between IgD and IgM is intrinsic to IgD and is independent of the experimental system or species (page 14, lines 293 – 295; page 17, lines 359 - 363).

4) Role of IgD in the germinal center response (Figure 7): The overall conclusion reached in this ms. that the germinal center response of IgD^{-/-} B cells is considerably delayed disagrees with results of a recent publication (Noviski et al., 2018). This could be due to experimental differences of various types. In the opinion of this reviewer, a more extensive analysis would be needed here to really provide useful insights, including a demonstration of a cell-intrinsic effect (e.g., using adoptive transfer or mixed bone marrow chimeric mice), since there could be cell extrinsic effects due to secreted antibody binding to the antigen and IgD deletion may alter the antibody secreted from the non-germinal center component of the antibody response.

Our data confirm previous findings reporting a delayed GC response in IgD^{-/-} mice (Roes and Rajewsky, 1993). Notably, immunization with sheep red blood cells (SRBC) (Noviski et al., 2018) mimics encounter with a potent and much more complex antigen than TNP-Ovalbumin, which was used in our study. Moreover, our concept suggests that B cells expressing either IgM alone or together with IgD show similar responses towards complex antigen. To test this directly, we immunized WT or IgD^{-/-} with SRBCs and analyzed the GC-response. In agreement with Noviski et al., we did not detect a significant difference in the development of the GC-response to SRBC in absence of IgD (Fig. S7).

Therefore, we conclude that efficient immune responses against pathogens with limiting antigen avidity might require IgD for directing B cells into GC reactions, while pathogens with strong complex antigen may not require IgD (page 16 - 17, lines 347 – 352; page 21, lines 434 – 438).

To test our finding in a competitive model, we prepared mixed bone marrow chimeras by transplanting mixtures of CD45.1⁺ WT and CD45.2⁺ IgD^{-/-} cells (see Fig. A below). However, we noticed that IgD^{-/-} B cells are underrepresented in the reconstituted

mice as compared to WT B cells at all tested ratios (see Fig. B-C below). As suggested by the literature, it is conceivable that the reduced amount of IgD^{-/-} B cells is a result of disturbed B cell homeostasis (Becker et al., 2017; Nitschke et al., 1993; Roes and Rajewsky, 1993; Sabouri et al., 2016). As the immunization of mixed bone marrow chimera represents a competitive model, this difference may affect the immunization results. In this case, it would be impossible to determine whether reduced numbers of GC IgD^{-/-} B cells are due to the absence of IgD or are caused by the fact that IgD^{-/-} B cells are underrepresented. Since the establishment of mixed bone marrow chimeras appears to be complicated by this phenomenon, extensive work is required to resolve this problem.

Figure for Reviewer #2, comment 4: Generation of WT/IgD^{-/-} bone marrow chimera (related to Fig. 7)

A | Schematic overview of the experimental design: CD45.1 and CD45.2 bone marrow cells were isolated from WT and IgD^{-/-} mice respectively, and lineage negative cells were purified by negative selection. CD45.1 and CD45.2 cells were mixed in a 1 : 2 ratio prior to transplantation into Rag^{-/-}/γC^{-/-} host animals.

B | Reconstitution was verified at 5 weeks after transplantation (as shown in A) by FACS analysis of the CD45.1/CD45.2 ratio in peripheral blood CD19⁺ B cells. The phenotype of injected cells was confirmed by IgM/IgD staining.

C | Percentages of CD45.1⁺ (black, bottom) and CD45.2⁺ (blue or green, top) cells in the blood of animals reconstituted with CD45.1⁺ WT cells and CD45.2⁺ WT or IgD^{-/-} B cells, respectively. Dots represent ratios from individual mice, mean. Cells were injected at ratios of either 1 : 1 or 1 : 2 (WT (CD45.1) : IgD^{-/-} (CD45.2)).

Minor comments:

1) The title of the manuscript overstates what can be concluded from the data presented (Pten deletion likely has effects on B cell responsiveness that go beyond just regulation of IgD expression).

The title does not exclude other roles that Pten likely has on B cell responsiveness.

2) Figure 4: the flow cytometry here should include appropriate negative control antibody staining profiles.

In the revised version of the manuscript, we included fluorescence minus one (FMO) staining as negative controls, which lack either the α -IgD or the α -CD23 antibody (page 41, lines 905 – 906).

3) Title to Figure 6 is an overstatement of what can be reasonably concluded here, these may be two distinct effects of Pten deficiency, rather than one being dependent on the other. Note also that Pten deficiency increases the calcium response to BCR crosslinking even in a non-energic setting, likely reflecting Btk and/or PLC-gamma2 binding to PIP3 in the plasma membrane as one parameter controlling their activation.

To make this point clearer, we added a quantification of IgM and IgD surface expression in mice of the respective genotypes (Fig. 6A). This quantification shows that in the presence of the cognate antigen, MD4^{tg} B cells down-regulate IgM expression leaving IgD as the pre-dominant BCR isotype (page 14 – 15, lines 303 – 307). As shown by previous data (Ubelhart et al., 2015) and in Fig. 5D & I, this results in reduced responsiveness to monovalent antigen. Upon inactivation of Pten, in MD4^{tg} B cells IgM expression remains comparable to that in WT cells whereas IgD expression is significantly reduced, thus altering the IgM/IgD ratio. We also included a quantification of the calcium measurements from Fig. 6B (Fig. 6C) showing that the Ca²⁺ response upon stimulation with α - κ LC antibodies is NOT enhanced in absence of Pten.

Referee #3:

Reviewer #3 evaluates our manuscript as a “solid study that addresses an interesting question”. In his/her opinion our data are “clearly presented and the experimentation and analysis are generally appropriate”.

Reviewer #3 however requests more work to clarify the discrepancy shown by the data in figure 7, concerning the role of IgD to recruit B cells into GCs and the study published by Zikherman et al. (Noviski eLife 2018).

*We thank the reviewer for her/his comments, which were addressed as described below. Changes in the manuscript text, referring to comments by reviewer #3 are marked in **blue**.*

Specific major concerns:

1) The authors show reduced IgD and CD23 in the absence of Pten and conclude that Pten regulates the generation of follicular B cells line 200-202. The alternative is that these markers are simply deregulated, which is a conclusion that may be supported by the regulation of splicing factors for IgD by FoxO1 and the relatively normal numbers of splenic B cells (figure 2E, S2). Some histology and staining of spleen and lymph nodes would test if the normal follicular structures are present.

This comment is also connected to point 1 raised by reviewer #2.

As suggested by the reviewer, we performed immunohistochemistry on spleen sections from $Pten^{ff}$, $Pten^{ff}$ x mb1-cre mice and $Pten$ -deficient mice with pre-rearranged Ig gene cassettes (Fig. 3E). Our data show abnormal follicles in the spleens of conditional $Pten$ -deficient mice. In contrast to spleens from WT mice, spleens from $Pten^{ff}$ x mb1-cre mice show no organized structures for B cells (Fig. 3E & S2) (Setz et al., 2018). Introducing pre-rearranged Ig genes increases the numbers of splenic B cells. However, the population of follicular B cells residing in the area between the marginal zone and T cell zone is smaller in the absence of $Pten$ as compared with WT controls. Therefore, we conclude that the development of mature Fo.B cells requires $Pten$ expression (page 10, lines 198 – 209; page 40, lines 890 - 895).

2) Figure 4 and S4. Figure S4 shows that *caFOXO1* expression results in more IgD mRNA versus controls. As IgD is a splice variant this analysis isn't really relevant as this could represent a change in IgH transcription in total. The key issue is to determine the proportion of total IgH transcripts that are IgD or IgM. In this context, it is also notable that FOXO loss (Figure 4E) appears to result in less surface IgM staining; this should also be quantified as a control.

Finally, I find the CD23 rescue unconvincing. The change in Figure 4B is very marginal and much below the WT situation. At the least CD23 mRNA expression here (and in figure 4E) should be measured using appropriate positive and negative controls as this phenomenon is predicted to be transcriptional.

IgD is not just expressed due to alternative splicing but also due to differential polyadenylation. The splicing factor Zfp318 has been described to promote IgD expression (Enders et al., 2014). If multiple poly-adenylation sites are present in a gene, several mRNAs can be generated that differ in their 3' end. Poly-adenylation site usage is regulated by poly-adenylation factors such as Cstf64 and E112. Both of which have been implicated in regulating the expression of the membrane-bound form of IgM (Martincic et al., 2009; Takagaki et al., 1996). It has been shown that presence of E112 promotes usage of the weaker proximal polyadenylation site that is used for example in plasma cells to generate soluble IgM. For generation of IgM BCR (equipped with the transmembrane domain), reduced levels of E112 are required. Reduction of E112 enables read-through to the non-consensus 5' splice site in the secretory-terminal exon and downstream membrane exons, thereby using the strong promoter-distal heavy-chain membrane poly-adenylation site. Since the poly-adenylation site used for production of IgD is located even further down-stream, we propose that E112 levels have to be decreased to achieve read-through to its localization. In line with this, we observed higher IgD surface expression upon FoxO1-A3 overexpression and down-regulation of E112.

The reduced IgM expression observed upon FoxO1 deletion (now included in Fig. S4G) may be attributed to reduced expression of *Cd79b* (Ig- β) (Dengler et al., 2008) (Fig. S4H). This hypothesis is further supported by our data showing that transcript levels of Ig- μ HC were not down-regulated following FoxO1 inactivation (Fig. S4H). Notably, measurement of CD79b and Ig- μ HC transcript levels represent appropriate

positive (CD79b) and negative (Ig- μ HC) controls, which we provided in the revised version of the manuscript (page 11, lines 232 – 237).

To determine whether expression of CD23 is also regulated by FoxO1 we included TaqMan-based analysis of *Fcer2* transcripts (encoding CD23) upon overexpression and deletion of FoxO1 (Fig. 4B&E, page 11 – 12, lines 237 – 239). In full agreement with the small increase in CD23 MFI upon FoxO1-A3 overexpression, we observed elevated *Fcer2* transcript levels. Moreover, deletion of FoxO1 leads to decreased CD23 MFI and to reduced *Fcer2* expression on mRNA level. These findings indicate that FoxO1 not only regulates IgD expression via Zfp318 and E12 but also expression of CD23. The small shift in CD23 surface expression was observed at 3 days after transduction. Therefore, it is conceivable that CD23 expression shows a lag phase after overexpression of FoxO1 and would probably increase more at later time points.

3) Figure 6. While the trends seem to fit with the authors' general model, the relatively small changes in Ca^{2+} flux need to be quantified over multiple experiments to determine how robust they are.

We repeated and quantified the Ca^{2+} measurements in 5 – 6 mice per genotype (Fig. 6C). Statistical analysis revealed that Ca^{2+} mobilization upon stimulation with soluble HEL is significantly higher in MD4^{tg} x ML5^{tg} double-transgenic mice lacking Pten compared to the same genotype on Pten-sufficient background (pages 43 – 44, lines 967 – 970).

We also included a quantification of IgM and IgD surface expression in B cells from mice of the respective genotypes to show more clearly how changes in the IgM/IgD ratio affect the responsiveness of B cells toward mono- and multivalent antigen (Fig. 6A).

4) Figure 7. I have several concerns about this data, particularly as it is important to the authors overall conclusion and disagrees with the results of SRBC immunization of $IgD^{-/-}$ mice shown in Noviski (eLife 2018). The GCB cell quantitation in panel also appears problematic, as the GC reaction seems both modest and transient, with a low peak at day 7. This appears to be a suboptimal response using NP-OVA in Alum. This experiment would be in WT much more convincing if the authors repeated the analysis and tracked the frequency of Ag specific NP-binding GCB cells, instead of the surrogate markers used.

This comment is related to point 4 raised by reviewer #2:

Our data confirm previous findings reporting a delayed GC response in $IgD^{-/-}$ mice (Roes and Rajewsky, 1993). Notably, Immunization with sheep red blood cells (SRBC) (Noviski et al., 2018) mimics encounter with a potent and much more complex antigen than TNP-Ovalbumin, which was used in our study. Moreover, our concept suggests that B cells expressing either IgM alone or together with IgD show similar responses towards complex antigen. To test this directly, we immunized WT or $IgD^{-/-}$ with SRBCs and analyzed the GC-response.

In agreement with Noviski et al., we did not detect a significant difference in the development of the GC-response to SRBC in absence of IgD (Fig. S7). Therefore, we conclude that efficient immune responses against pathogens with limiting antigen avidity might require IgD for directing B cells into GC reactions, while pathogens with strong complex antigen may not require IgD (page 16 - 17, lines 347 – 352; page 21, lines 434 – 438).

As we detected antigen-specific IgG in the sera of immunized WT mice already at 7 days after immunization (Fig. 7G), this suggests that there was an efficient response to TNP-Ova. Moreover, we detected higher levels of TNP-specific IgG in sera of WT mice as compared with $IgD^{-/-}$ mice indicating that the delayed GC reaction also affects the humoral immune response. This finding also confirms that the observed delay in the GC response is not just based on surrogate markers but also results in reduced production of class-switched antigen-specific antibodies (page 16, lines 344 – 346; page 45, lines 1000 - 1005).

Together, we assume that this specific function of IgD might become evident only upon encounter with weak immunogens or pathogens with limiting antigen avidity.

Notably, the prevailing conditions during an infection might also be “suboptimal” for GC responses suggesting that IgD fulfils an important function in protection from pathogens.

References

- Barthel, A., Schmoll, D., and Unterman, T.G. (2005). FoxO proteins in insulin action and metabolism. *Trends Endocrinol Metab* *16*, 183-189.
- Becker, M., Hobeika, E., Jumaa, H., Reth, M., and Maity, P.C. (2017). CXCR4 signaling and function require the expression of the IgD-class B-cell antigen receptor. *Proc Natl Acad Sci U S A* *114*, 5231-5236.
- Dengler, H.S., Baracho, G.V., Omori, S.A., Bruckner, S., Arden, K.C., Castrillon, D.H., DePinho, R.A., and Rickert, R.C. (2008). Distinct functions for the transcription factor Foxo1 at various stages of B cell differentiation. *Nat Immunol* *9*, 1388-1398.
- Enders, A., Short, A., Miosge, L.A., Bergmann, H., Sontani, Y., Bertram, E.M., Whittle, B., Balakishnan, B., Yoshida, K., Sjollem, G., *et al.* (2014). Zinc-finger protein ZFP318 is essential for expression of IgD, the alternatively spliced Igh product made by mature B lymphocytes. *Proc Natl Acad Sci U S A* *111*, 4513-4518.
- Furuyama, T., Nakazawa, T., Nakano, I., and Mori, N. (2000). Identification of the differential distribution patterns of mRNAs and consensus binding sequences for mouse DAF-16 homologues. *Biochem J* *349*, 629-634.
- Halverson, R., Torres, R.M., and Pelanda, R. (2004). Receptor editing is the main mechanism of B cell tolerance toward membrane antigens. *Nat Immunol* *5*, 645-650.
- Kim, Y.M., Pan, J.Y., Korbel, G.A., Peperzak, V., Boes, M., and Ploegh, H.L. (2006). Monovalent ligation of the B cell receptor induces receptor activation but fails to promote antigen presentation. *Proc Natl Acad Sci U S A* *103*, 3327-3332.
- Lin, Y.C., Jhunjhunwala, S., Benner, C., Heinz, S., Welinder, E., Mansson, R., Sigvardsson, M., Hagman, J., Espinoza, C.A., Dutkowski, J., *et al.* (2010). A global network of transcription factors, involving E2A, EBF1 and Foxo1, that orchestrates B cell fate. *Nat Immunol* *11*, 635-643.
- Martincic, K., Alkan, S.A., Cheatle, A., Borghesi, L., and Milcarek, C. (2009). Transcription elongation factor ELL2 directs immunoglobulin secretion in plasma cells by stimulating altered RNA processing. *Nat Immunol* *10*, 1102-1109.
- Nitschke, L., Kosco, M.H., Kohler, G., and Lamers, M.C. (1993). Immunoglobulin D-deficient mice can mount normal immune responses to thymus-independent and -dependent antigens. *Proc Natl Acad Sci U S A* *90*, 1887-1891.
- Noviski, M., Mueller, J.L., Satterthwaite, A., Garrett-Sinha, L.A., Brombacher, F., and Zikherman, J. (2018). IgM and IgD B cell receptors differentially respond to endogenous antigens and control B cell fate. *Elife* *7*.
- Roes, J., and Rajewsky, K. (1993). Immunoglobulin D (IgD)-deficient mice reveal an auxiliary receptor function for IgD in antigen-mediated recruitment of B cells. *J Exp Med* *177*, 45-55.

Sabouri, Z., Perotti, S., Spierings, E., Humburg, P., Yabas, M., Bergmann, H., Horikawa, K., Roots, C., Lambe, S., Young, C., *et al.* (2016). IgD attenuates the IgM-induced anergy response in transitional and mature B cells. *Nat Commun* 7, 13381.

Setz, C.S., Hug, E., Khadour, A., Abdelrasoul, H., Bilal, M., Hobeika, E., and Jumaa, H. (2018). PI3K-Mediated Blimp-1 Activation Controls B Cell Selection and Homeostasis. *Cell Rep* 24, 391-405.

Shin, D.J., Joshi, P., Hong, S.H., Mosure, K., Shin, D.G., and Osborne, T.F. (2012). Genome-wide analysis of FoxO1 binding in hepatic chromatin: potential involvement of FoxO1 in linking retinoid signaling to hepatic gluconeogenesis. *Nucleic Acids Res* 40, 11499-11509.

Srinivasan, L., Sasaki, Y., Calado, D.P., Zhang, B., Paik, J.H., DePinho, R.A., Kutok, J.L., Kearney, J.F., Otipoby, K.L., and Rajewsky, K. (2009). PI3 kinase signals BCR-dependent mature B cell survival. *Cell* 139, 573-586.

Takagaki, Y., Seipelt, R.L., Peterson, M.L., and Manley, J.L. (1996). The polyadenylation factor CstF-64 regulates alternative processing of IgM heavy chain pre-mRNA during B cell differentiation. *Cell* 87, 941-952.

Ubelhart, R., Hug, E., Bach, M.P., Wossning, T., Duhren-von Minden, M., Horn, A.H., Tsiantoulas, D., Kometani, K., Kurosaki, T., Binder, C.J., *et al.* (2015). Responsiveness of B cells is regulated by the hinge region of IgD. *Nat Immunol* 16, 534-543.

Thank you for submitting the revised manuscript to The EMBO Journal. I am sorry for the delay in getting back to you with a decision, but I have now received the comments back from the referees on the paper.

As you can see below, the referees appreciate the introduced changes and support publication here. Referee # 2 still has some remaining concerns, but I think they can all be addressed with appropriate text changes. Let me know if we need to discuss any of them further.

When you submit your revised manuscript would you also take care of the following things:

 REFEREE REPORTS:

Referee #1:

My comments have been adequately addressed by the authors. I have no additional comments and feel that this study is now ready for publication. I plaud the authors for this elegant and novel work.

Referee #2:

The authors have added new data to improve their manuscript.

The authors' descriptions of what can be concluded from their experimental evidence remain problematic, in the opinion of this reviewer, for the reasons described below and therefore the text of the ms. should be edited to be more straightforward and accurate in interpreting the data presented.

1) The authors state at the end of the Abstract and elsewhere (first paragraph of Discussion, etc.) that "... IgD expression results in B cells that are selectively responsive to antigen complexes and are thus capable of initiating efficient T cell dependent antibody responses". This statement strikes me as misleading (or not easily understood) in two ways. Firstly, a simple reading of this statement strikes me as being contradicted by the data presented on GC responses in Fig. 7 and Supp Fig. S7. The authors have added data with SRBCs (clearly a complex antigen), showing that the GC response to this complex antigen is more or less OK in IgD^{-/-} mice, whereas GC responses to other, simpler antigens are delayed in IgD^{-/-} mice compared to the responses of wild type mice. This does not match well to the statement quoted above. Also, these results seem to match poorly with the expectations from the signaling experiments. The mechanistic signaling studies show that what is unique about IgD vs. IgM BCRs is that only IgM BCR can signal well to monomeric antigen, but BOTH can signal in response to higher valency antigens, so logically, IgD would contribute more to the response to antigen complexes and would have little impact the response to low valency, soluble antigens, or might inhibit such responses (since if IgD is present, it could inhibit signaling from IgM by competing for binding to antigen; see Fig 5E). The results seem opposite to these expectations. Adding the SRBC GC data to the ms. is good as it helps reconcile their results with previously published data, but the final sentence of the Abstract is contradictory or easily misunderstood, as are other parts of the text. Clearly, the text related to this issue should be re-written to address what appear to be results that contradict the authors' model.

2) PTEN-deficiency of B cells and consequent loss of FoxO expression (or direct deletion of FoxO1) leads to a loss of expression of surface IgD, and this is shown clearly, but two interpretations are alternatively embraced, rather than discussed in a clear and informative manner. PTEN-deficiency may block maturation from splenic T1 stage to FO B cells (which includes upregulation of IgD) OR there could be a direct role for FoxO in surface IgD expression OR both may be true (in other words, FoxO regulates a series of genes that regulate IgD expression AND other aspects of maturation from T1 to FO stage). The authors go back and forth about this without giving a clear explanation.

3) Moreover, related to the previous point, on p. 10 lines 208-209, the authors evidently conclude that IgD expression controls maturation from T1 to FO ("these results suggest that PTEN interferes with IgD expression thereby altering their developmental fate"). Actually this is backwards, deletion of PTEN (not PTEN function) interferes with IgD expression, but in addition, what is the reason for saying that IgD expression "alters" developmental fate (e.g. T1 to FO maturation)? Is there a

maturational arrest in IgD^{-/-} B cells? The discussion should address this issue as it relates to what we can learn from studying PTEN-deficient or FoxO-deficient B cells expressing IgH + IgL transgenes (to overcome the lack of VDJ recombination in the absence of FoxO) and how we should interpret those results.

4) The discussion, of "anergic B cells" is, to this reviewer's thinking, skewed to an incorrect definition and therefore logically flawed. Anergy of B cells was first described by Gus Nossal and coworkers who demonstrated that tolerized antigen-specific B cells could be viable but unresponsive in terms of making antibody responses. Chris Goodnow showed the same phenomenon with an Ig transgenic system (the MD4 Ig transgenic, used also in this ms.). Subsequently, they found that anergic B cells are competent to participate in GC responses. Other manifestations observed by Goodnow and subsequent investigators, including downregulated surface IgM and reduced BCR signaling in response to low valency antigen, are likely relevant, but are not the key defining parameter. In particular, simply measuring calcium responses is an inadequate measure. In Fig. 3D, the authors present anti-HEL IgG data, which shows that MD4 x ML5 mice produce very low anti-HEL antibody titers, unlike MD4 mice (orange triangles vs. red squares), which is the standard definition of anergy, and moreover that MD4 or MD4 x ML5 mice with PTEN deletion in B cells produce even higher titers of anti-HEL, demonstrating that anergy of MD4 x ML5 mice is broken by deletion of PTEN in B cells.

5) The Discussion has several particular claims about B cell anergy that seem to be inaccurate: Discussion line 404 "argues against an essential role of PTEN in anergy" : Fig 3D shows it is essential in the MD4 system. It may not be essential in the 3-83 Ig transgenic system, in which receptor editing is the main tolerance mechanism; Discussion line 408 "MD4 x ML5 B cells are not anergic ...". This is clearly incorrect as they have been shown to have diminished antibody responses when provided with T cell help (Goodnow's publications, reproduced in Fig. 3D). Moreover, Browne et al showed that the Akt response to Fab'2 anti-kappa was strongly attenuated, so it is not the case that IgD signals normally in these B cells. Discussion lines 410-412, "B cells that downregulate both IgM and IgD are the truly anergic B cells": This claim seems to be based on 3-83 Ig transgenic x PTEN deleted situation, which is obviously somewhat artificial (PTEN deficiency probably keeps these cells alive, they would otherwise die rapidly). Most people in this field tend to think that the IgM^{low} IgD^{high} follicular B cells represent anergic B cells. The data in Fig. 3D show that the MD4 x ML5 B cells exhibit anergy, as does a lot of published work. The data in Fig. 6C are consistent with reduced calcium signaling of MD4 x ML5 B cells to anti-kappa stimulation, although there is scatter in the data, so the data in this ms. is equivocal, but note in Browne et al, the showed clearly impaired Akt activation to anti-kappa stimulation. Discussion lines 413-415, "anergic B cells can only be detected when receptor editing and clonal deletion are blocked"; this is true for 3-83 Ig transgenic, but not other Ig transgenic systems which generate anergic B cells, of which there are several (for example, in addition to MD4, some of the anti-DNA Ig transgenics and the anti-ARS Ig transgenic studied by Cambier's group). Discussion line 467-468 "referring to IgD expressing cells as anergic confuses anergy with maturation"; for the reasons stated above the MD4 x ML5 Ig transgenic system represents a counter-example, as do others, such as the Jan Erikson 3H9 V_λ Ig transgenic system (anti-dsDNA). The authors have a radical view of B cell anergy that is in conflict with numerous other investigators in the field and also is contradicted by the authors data, particularly in Fig. 3D.

6) I still think the Title is a poor description of what has been shown in this study. In particular, the authors have not examined GC responses of their PTEN^{fl/fl} x mb1-cre mice, but rather are depending on previously published work. If, for example, IgD deletion has a much smaller effect on the GC response than PTEN deletion, then the title would clearly be misleading. Moreover, the discussion largely ignores the conclusion of the title, but rather focuses on other issues such as the role of PTEN and IgD in anergy.

Minor comments

1) Fig 4E, label for lowest FACS profile on right side, should this be "Cre-GFP" instead of "FoxO1-A3"?

Referee #3:

The authors have provided some interesting new data and adequately addressed my concerns.

Point by point response

Reviewer #2 asked for additional discussion of the data presented in the manuscript:

1) The authors state at the end of the Abstract and elsewhere (first paragraph of Discussion, etc.) that "... IgD expression results in B cells that are selectively responsive to antigen complexes and are thus capable of initiating efficient T cell dependent antibody responses". This statement strikes me as misleading (or not easily understood) in two ways. Firstly, a simple reading of this statement strikes me as being contradicted by the data presented on GC responses in Fig. 7 and Supp Fig. S7. The authors have added data with SRBCs (clearly a complex antigen), showing that the GC response to this complex antigen is more or less OK in IgD^{-/-} mice, whereas GC responses to other, simpler antigens are delayed in IgD^{-/-} mice compared to the responses of wild type mice. This does not match well to the statement quoted above. Also, these results seem to match poorly with the expectations from the signaling experiments. The mechanistic signaling studies show that what is unique about IgD vs. IgM BCRs is that only IgM BCR can signal well to monomeric antigen, but BOTH can signal in response to higher valency antigens, so logically, IgD would contribute more to the response to antigen complexes and would have little impact the response to low valency, soluble antigens, or might inhibit such responses (since if IgD is present, it could inhibit signaling from IgM by competing for binding to antigen; see Fig 5E). The results seem opposite to these expectations. Adding the SRBC GC data to the ms. is good as it helps reconcile their results with previously published data, but the final sentence of the Abstract is contradictory or easily misunderstood, as are other parts of the text. Clearly, the text related to this issue should be re-written to address what appear to be results that contradict the authors' model.

The role of IgD in GC reaction is concluded from our data showing that the GC reaction after TNP-Ova immunization is delayed in IgD-deficient mice. This observation is in agreement with previous data showing that IgD is required for efficient T-cell dependent immune responses (Roes and Rajewsky, 1993). Notably, TNP-Ova represents a multivalent antigen consisting of 10 - 20 TNP molecules coupled to a single Ova molecule. Moreover, B cell activation by SRBCs has been

suggested to be mediated via pattern recognition receptors and thus to be BCR-independent (Loetsch et al., 2017). In fact, immunization with SRBCs does not require any additional adjuvants while immunization with TNP-Ova does. In line with this, B cells from both *MD4* single- and *MD4 x ML5* double-transgenic mice have been shown to secrete comparable amounts of HEL-specific IgM antibodies upon stimulation with the TLR9 ligand CpG (Figure 2C in Rui et al., 2003). The finding that IgD-deficient mice show normal immune responses after immunization with SRBCs suggests that a BCR-independent “superantigen” is capable of activating the compromised IgD-deficient B cells thereby masking the actual role of IgD for the immune system. In full agreement, *Pten*-deficient B cells mount a strong GC response after immunization with SRBCs.

Therefore, we concluded that efficient immune responses against pathogens with limiting antigen avidity may require IgD for rapid recruitment of B cells into T-dependent immune responses, while stimulation via IgM primarily leads to plasma cell differentiation and IgM secretion. The limiting antigen still has to be multivalent for IgD to activate the respective B cells. Thus, the immunization experiments with TNP-Ova and SRBCs are in line with the overall concept of this manuscript.

To further clarify this point, we modified the discussion and included additional passages in the revised version of the manuscript (page 17, lines 336 – 338; page 17 – 18, lines 353 – 356; page 23, lines 457 – 473). Importantly, to avoid generalization, we repetitively specify in the context of GC responses which antigen was used to draw the included conclusions.

2) PTEN-deficiency of B cells and consequent loss of FoxO expression (or direct deletion of FoxO1) leads to a loss of expression of surface IgD, and this is shown clearly, but two interpretations are alternatively embraced, rather than discussed in a clear and informative manner. PTEN-deficiency may block maturation from splenic T1 stage to FO B cells (which includes upregulation of IgD) OR there could be a direct role for FoxO in surface IgD expression OR both may be true (in other words, FoxO regulates a series of genes that regulate IgD expression AND other aspects of maturation from T1 to FO stage). What is the reason for saying that IgD expression "alters" developmental fate (e.g. T1 to FO maturation)? Is there a maturational arrest in IgD^{-/-} B cells? The discussion should address this issue as it relates to what we can learn from studying PTEN-deficient or FoxO-deficient B cells expressing IgH +

IgL transgenes (to overcome the lack of VDJ recombination in the absence of FoxO) and how we should interpret those results.

As already suggested by reviewer #3 in the first round of reviewing, we performed immunohistochemistry on spleen sections from *Pten^{ff}*, *Pten^{ff}* x mb1-cre mice and *Pten*-deficient mice with pre-rearranged *Ig* gene cassettes (Fig. 3E). Our data show abnormal follicles in the spleens of conditional *Pten*-deficient mice. In contrast to spleens from WT mice, spleens from *Pten^{ff}* x mb1-cre mice show no organized structures for B cells (Fig. 3E; Appendix Fig. S2; page 11 lines 201 – 209 and (Setz et al., 2018)). Introducing pre-rearranged *Ig* genes increases the numbers of splenic B cells. Still, the population of follicular B cells is smaller in the absence of *Pten* as compared with WT controls. Therefore, we concluded that the development of Fo.B cells is affected in the absence of *Pten*/*FoxO1* function in addition to the impaired expression of Fo.B cell markers such as IgD and CD23. In the revised version of the manuscript, we pointed out that B cells might be blocked at the T1 in the absence of *Pten*/*FoxO1* function (page 11, lines 209 – 212).

3) The discussion, of "anergic B cells" is, to this reviewer's thinking, skewed to an incorrect definition and therefore logically flawed. Anergy of B cells was first described by Gus Nossal and coworkers who demonstrated that tolerized antigen-specific B cells could be viable but unresponsive in terms of making antibody responses. Chris Goodnow showed the same phenomenon with an Ig transgenic system (the MD4 Ig transgenic, used also in this ms). Subsequently, they found that anergic B cells are competent to participate in GC responses. Other manifestations observed by Goodnow and subsequent investigators, including downregulated surface IgM and reduced BCR signaling in response to low valency antigen, are likely relevant, but are not the key defining parameter. In particular, simply measuring calcium responses is an inadequate measure.

In Fig. 3D, the authors present anti-HEL IgG data, which shows that MD4 x ML5 mice produce very low anti-HEL antibody titers, unlike MD4 mice (orange triangles vs. red squares), which is the standard definition of anergy, and moreover that MD4 or MD4 x ML5 mice with PTEN deletion in B cells produce even higher titers of anti-HEL, demonstrating that anergy of MD4 x ML5 mice is broken by deletion of PTEN in B cells.

As stated by the reviewer, B cells from *MD4*-transgenic mice, generated by C. Goodnow and used in our study, were proposed to be anergic when crossed to *ML5*-transgenic mice to introduce the antigen. Importantly, the “anergic B cells” in the double-transgenic mice were initially described as being functionally unresponsive. Accordingly, anergic B cells are not functional. Later, as also stated by this reviewer, it was reported that anergic B cells are functional and participate in GC responses. Accordingly, anergic B cells are functionally competent. Since this is inconsistent, we provide an alternative concept based on published literature and our data. Our concept defines anergy as functional unresponsiveness. Since B cells from *MD4* x *ML5* double-transgenic mice are competent to participate in GC responses (Sabouri et al., 2014; Sabouri et al., 2016), induce robust signaling to stimulation with anti-BCR antibodies (Cooke et al., 1994) and efficiently respond to multivalent antigen (Ubelhart et al., 2015), we think it is not logical to describe them as anergic B cells. Since B cells from *MD4* x *ML5* double-transgenic mice resemble mature splenic B cells from wild type mice, showing high IgD and reduced IgM expression, in addition to the ability to induce signaling as well as being immune competent, it is more logical to describe these cells as normal mature B cells.

Recently, the scientists who proposed B cell anergy in *MD4* x *ML5* double-transgenic mice have introduced the term “mature anergic B cells” which, according to our view, confuses two fundamentally different conditions. Therefore, a major point put forward by our manuscript is the discrimination between immune competent mature B cells, characterized by modulated surface BCR expression (IgM^{low}/IgD^{high} on mature wild type B cells or on *MD4* x *ML5* double-transgenic B cells), and real anergic B cells, which completely down-regulate surface BCR expression and are thus functionally unresponsive (see Figures 2A-B & 3A). We believe that a clear distinction between mature B cells and anergic cells is important for understanding B cell development and antibody responses.

The last paragraph of point #3 raised by the reviewer (marked in gray) is related to point #4 and discussed below.

4) The Discussion has several particular claims about B cell anergy that seem to be inaccurate: Discussion line 404 "argues against an essential role of PTEN in anergy": Fig 3D shows it is essential in the *MD4* system. It may not be essential in the 3-83 Ig

transgenic system, in which receptor editing is the main tolerance mechanism; Discussion line 408 "MD4 x ML5 B cells are not anergic...". This is clearly incorrect as they have been shown to have diminished antibody responses when provided with T cell help (Goodnow's publications, reproduced in Fig. 3D).

Moreover, Browne et al showed that the Akt response to Fab'2 anti-kappa was strongly attenuated, so it is not the case that IgD signals normally in these B cells. Discussion lines 410-412, "B cells that downregulate both IgM and IgD are the truly anergic B cells": This claim seems to be based on 3-83 Ig transgenic x PTEN deleted situation, which is obviously somewhat artificial (PTEN deficiency probably keeps these cells alive, they would otherwise die rapidly). Most people in this field tend to think that the IgM^{low} IgD^{high} follicular B cells represent anergic B cells. The data in Fig. 3D show that the MD4 x ML5 B cells exhibit anergy, as does a lot of published work. The data in Fig. 6C are consistent with reduced calcium signaling of MD4 x ML5 B cells to anti-kappa stimulation, although there is scatter in the data, so the data in this ms. is equivocal, but note in Browne et al, they showed clearly impaired Akt activation to anti-kappa stimulation. Discussion lines 413-415, "anergic B cells can only be detected when receptor editing and clonal deletion are blocked"; this is true for 3-83 Ig transgenic, but not other Ig transgenic systems which generate anergic B cells, of which there are several (for example, in addition to MD4, some of the anti-DNA Ig transgenics and the anti-ARS Ig transgenic studied by Cambier's group). Discussion line 467-468 "referring to IgD expressing cells as anergic confuses anergy with maturation"; for the reasons stated above the MD4 x ML5 Ig transgenic system represents a counter-example, as do others, such as the Jan Erikson 3H9 V λ Ig transgenic system (anti-dsDNA). The authors have a radical view of B cell anergy that is in conflict with numerous other investigators in the field and also is contradicted by the authors data, particularly in Fig. 3D.

Using the 3-83^{ki} transgenic model, with conditional Pten deletion on the autoreactive background, we detect B cells that (i) lack surface BCR-expression, that (ii) neither respond to external stimulation and (iii) nor differentiate into antibody-secreting cells, which is in line with the definition of anergy by Gus Nossal and coworkers cited by this reviewer.

The 3-83 model unambiguously shows that Pten is dispensable for anergy induction (i.e. the generation of real functionally unresponsive cells). Consequently, general

statements about an essential role of Pten in B cell energy (Browne et al., 2009) are not supported.

In contrast to (Browne et al., 2009), our findings reveal no significant differences in calcium signaling intensity between B cells from *MD4*-single-transgenic and *MD4* x *ML5*-double-transgenic mice upon stimulation with anti-kappa antibodies (Fig. 6C and (Ubelhart et al., 2015)). The scatter of the data points depicted in the quantification of our manuscript (Fig. 6C) is based on multiple measurements, which is a sign of quality and robustness of our findings. Browne et al. neither show multiple measurements for their data (Ca^{2+} mobilization and Akt phosphorylation upon anti-IgM or anti-kappa stimulation) nor quantification or statistics and therefore their data cannot be compared with those in our study.

To further support our conclusion, we repeated the Ca^{2+} measurements with cells from 3 individual *MD4* single- and *MD4* x *ML5* double-transgenic mice. The results are in full agreement with our previous findings and show that there is no significant difference in responsiveness between B cells from these two groups of mice:

In *MD4* single-transgenic mice, in absence of antigen, B cells do not complete their development to become mature ($\text{IgM}^{\text{low}}/\text{IgD}^{\text{high}}$) B cells. Since these cells secrete

HEL-specific IgM spontaneously, this antibody secretion is induced by antigen-independent mechanisms. B cells from *MD4 x ML5* double-transgenic mice become mature B cells that downregulate IgM, express mainly IgD and therefore require multivalent antigen for activation and secretion of HEL-specific antibodies. Similarly, mature B cells from wild type mice do not secrete IgD, neither spontaneously nor after activation. This is in agreement with our concept that B cells from *MD4 x ML5* double-transgenic mice resemble mature B cells.

It has previously been shown that Blimp-1 expression is deregulated in the absence of Pten (Omori et al., 2006), thereby promoting terminal differentiation in immature B cells and leading to increased plasma cell differentiation and antibody secretion (Setz et al., 2018). Thus, increased IgM concentrations in the absence of Pten cannot simply be interpreted as result of defective tolerance induction. The increased concentration of serum IgM is induced by elevated spontaneous antibody secretion, which results from deregulated Blimp-1 expression as consequence of Pten inactivation.

We did not test whether Pten-deficiency may affect the anergy that was reported using the transgenic mice indicated by the reviewer. It should be noted, however, that most systems use classical transgenes, interfering with normal IgD expression. As requested by the reviewer, we addressed in the discussion of the revised version the idea that additional animal models for anergy exist and that deleting Pten in these models may be useful for further characterization of the role of Pten in B cell responsiveness (page 21, lines 417 - 421).

5) I still think the title is a poor description of what has been shown in this study. In particular, the authors have not examined GC responses of their *PTEN^{fl/fl} x mb1-cre* mice, but rather are depending on previously published work. If, for example, IgD deletion has a much smaller effect on the GC response than *PTEN* deletion, then the title would clearly be misleading. Moreover, the discussion largely ignores the conclusion of the title, but rather focuses on other issues such as the role of *PTEN* and IgD in anergy.

We selected the title because (i) our data show that Pten activates IgD expression via FoxO1, (ii) IgD expressing B cells show selective responsiveness to multivalent antigen and because (iii) IgD-deficient mice, compared with wild type mice, show a

delayed GC reaction to TNP-Ova which is in agreement with previous reports suggesting that efficient T-cell dependent immune responses require IgD (Roes and Rajewsky, 1993).

Moreover, the role of Pten/FoxO1 in GC reaction has been reported previously and the related literature is briefly described in the introduction of our manuscript (page 5, lines 87 – 91). Altogether, it is allowed to conclude that IgD regulates B cell responsiveness, plays a role in GC reaction and is thereby regulated by Pten. Importantly, the title does not exclude the involvement of additional Pten/FoxO1-regulated factors in these processes.

As requested by the reviewer, the title of the manuscript has now been discussed in the revised manuscript (page 19, lines 361 – 365).

Moreover, we immunized Pten-deficient mice with SRBCs and found that they mount efficient GC responses. So, we can exclude that Pten-deficient mice are less efficient than IgD-KO mice in response to immunization with SRBCs.

Minor comments:

1) Fig 4E, label for lowest FACS profile on right side, should this be "Cre-GFP" instead of "FoxO1-A3"?

We have corrected this labeling mistake.

References

Browne, C.D., Del Nagro, C.J., Cato, M.H., Dengler, H.S., and Rickert, R.C. (2009). Suppression of phosphatidylinositol 3,4,5-trisphosphate production is a key determinant of B cell anergy. *Immunity* 31, 749-760.

Cooke, M.P., Heath, A.W., Shokat, K.M., Zeng, Y., Finkelman, F.D., Linsley, P.S., Howard, M., and Goodnow, C.C. (1994). Immunoglobulin signal transduction guides the specificity of B cell-T cell interactions and is blocked in tolerant self-reactive B cells. *J Exp Med* 179, 425-438.

Loetsch, C., Warren, J., Laskowski, A., Vazquez-Lombardi, R., Jandl, C., Langley, D.B., Christ, D., Thorburn, D.R., Ryugo, D.K., Sprent, J., Batten, M., and King, C. (2017). Cytosolic Recognition of RNA Drives the Immune Response to Heterologous Erythrocytes. *Cell Rep* 21, 1624-1638.

Omori, S.A., Cato, M.H., Anzelon-Mills, A., Puri, K.D., Shapiro-Shelef, M., Calame, K., and Rickert, R.C. (2006). Regulation of class-switch recombination and plasma cell differentiation by phosphatidylinositol 3-kinase signaling. *Immunity* 25, 545-557.

Roes, J., and Rajewsky, K. (1993). Immunoglobulin D (IgD)-deficient mice reveal an auxiliary receptor function for IgD in antigen-mediated recruitment of B cells. *J Exp Med* 177, 45-55.

Rui, L., Vinuesa, C.G., Blasioli, J., and Goodnow, C.C. (2003). Resistance to CpG DNA-induced autoimmunity through tolerogenic B cell antigen receptor ERK signaling. *Nat Immunol* 4, 594-600.

Sabouri, Z., Perotti, S., Spierings, E., Humburg, P., Yabas, M., Bergmann, H., Horikawa, K., Roots, C., Lambe, S., Young, C., Andrews, T.D., Field, M., Enders, A., Reed, J.H., and Goodnow, C.C. (2016). IgD attenuates the IgM-induced anergy response in transitional and mature B cells. *Nat Commun* 7, 13381.

Sabouri, Z., Schofield, P., Horikawa, K., Spierings, E., Kipling, D., Randall, K.L., Langley, D., Roome, B., Vazquez-Lombardi, R., Rouet, R., Hermes, J., Chan, T.D., Brink, R., Dunn-Walters, D.K., Christ, D., and Goodnow, C.C. (2014). Redemption of autoantibodies on anergic B cells by variable-region glycosylation and mutation away from self-reactivity. *Proc Natl Acad Sci U S A* 111, E2567-2575.

Setz, C.S., Hug, E., Khadour, A., Abdelrasoul, H., Bilal, M., Hobeika, E., and Jumaa, H. (2018). PI3K-Mediated Blimp-1 Activation Controls B Cell Selection and Homeostasis. *Cell Rep* 24, 391-405.

Ubelhart, R., Hug, E., Bach, M.P., Wossning, T., Duhren-von Minden, M., Horn, A.H., Tsiantoulas, D., Kometani, K., Kurosaki, T., Binder, C.J., Sticht, H., Nitschke, L., Reth, M., and Jumaa, H. (2015). Responsiveness of B cells is regulated by the hinge region of IgD. *Nat Immunol* 16, 534-543.

3rd Editorial Decision

21st Mar 2019

Thanks for sending us your revised manuscript. I have now looked at everything and all looks good. There are just some editorial things we have to sort out with the files.

That should be all - once I get the revised version back in then I will send the formal acceptance letter.

Corresponding Author Name: Hassan Jumaa

Journal Submitted to: EMBO J

Manuscript Number: EMBOJ-2018-100249